# Loss of Pol III repressor Maf1 in neurons promotes longevity by preventing the age-related decline in 5S rRNA and translation

**Bowen Xu, Alexander Hull, Olivia N. M. Hill, Naja Kobal, Enric Ureña, Linda Partridge**\*, **Nazif Alic**⬤\*

 Department of Genetics, Evolution and Environment, Institute of Healthy Ageing, University College London, London, United Kingdom

\* linda.partridge@ucl.ac.uk (LP); n.alic@ucl.ac.uk (NA)

## Abstract

Attenuating protein synthesis promotes longevity in multiple species. However, numerous studies indicate that aging drives a decrease in protein synthetic capacity. These observations hint at potential, unexplored benefits of stimulating protein synthesis in old age. In this work, we focus on Maf1, a repressor of protein synthesis genes transcribed by RNA Polymerase (Pol) III, such as the 5S rRNA and tRNAs, and its role in aging. We show that the knockdown of Maf1 extends lifespan in *Drosophila.* Maf1 limits longevity specifically from adult neurons in both female and male fruit flies. In older females, adult neuron-specific knockdown of Maf1 improves neuromuscular function as well as the function of a distal organ, the gut. We find that the extension of female lifespan upon Maf1 knockdown requires Pol III initiation on the 5S rRNA. Indeed, reducing neuronal Maf1 activity rescues the age-related decline in 5S expression and protein synthesis in the brain of female flies. Hence, our findings show that stimulating neuronal protein synthesis can promote healthy aging.

## Introduction

The proportion of older people is steadily increasing in many parts of the world, making it important to uncover new innovations to improve health into older age [1]. The rate of protein synthesis naturally declines in many tissues during aging. This decline has been extensively documented in different tissues and cell types of multiple species, including humans [2–4]. While this decline may be beneficial in certain cell types, [5], it is also likely to compromise cellular function, especially in cell types such as neurons, where de novo protein synthesis underpins crucial cellular tasks [3,6,7], thus contributing to organismal aging. However, the potential health benefits of increasing the capacity for protein synthesis during aging, to combat this natural decline, have not received much attention. Rather, the focus of research into the role of protein synthesis in aging has been on the longevity-promoting effects of reduced

**Data availability statement:** The authors confirm that all data underlying the findings are fully available without restriction. All relevant data are within the paper and its Supporting information files.

**Funding:** The work was funded in part by Biotechnology and Biological Sciences Research Council (BBSRC) grant BB/S014357/1 to NA, BBSRC grant BB/V006541/1 to LP, Medical research Council and BBSRC BLAST Ageing Research network's funding BB/W01825X/1 to BW and Genetic Society Summer Studentship to ONMH. The funders had no role in study design, data collection and analysis, decision to publish, or preparation of the manuscript.

**Competing interests:** The authors have declared that no competing interests exist.

**Abbreviations:** ALS, amyotrophic lateral sclerosis; FTD, frontotemporal dementia; Pol, Polymerase; RNAi, RNA interference; TAGs, triacylglycerols; TFs, transcription factors.

translation: decreased provision of ribosomal components or translation initiation factors, as well as pharmacological inhibition of protein synthesis, and in fruit flies this is specifically relevant in early adulthood, can all boost late-life health and extend lifespan across animal models [4,5,8–11], with evidence of relevance to mammals including humans [12,13]. The underlying mechanisms are not fully elucidated but are likely to be indirect, i.e., not resulting from decreased protein synthesis per se. Recent research has started extending this narrow focus: increasing the fidelity of translation directly has also been shown to promote longevity [14]. However, and despite the decades of evidence for an age-related decline in translation capacity, the longevity-promoting potential of increased translation remains unexplored.

Maf1 is an evolutionarily conserved, global repressor of RNA polymerase (Pol) III [15], which transcribes short, non-coding RNAs including the 5S rRNA and tRNAs required for protein translation [16–18]. Initially implicated in nutrient and stress responses in budding yeast [19], the role of Maf1 in animal development and physiology is becoming increasingly appreciated [15,20]. For example, in fruit flies, *dMaf1* limits organismal growth [21], while in mice *Maf1* restricts futile, energetically costly cycles of tRNA synthesis and degradation [22–24], regulates adipocyte and osteoblast differentiation and reduces cardiac hypertrophy [25–27]. Specifically in neurons, Maf1 limits dendritic outgrowth, axonal regeneration and supresses neural repair [27–30]. Interestingly, mice lacking *Maf1* activity have several improved health parameters and females are long-lived [22,23], a phenotype similarly observed in *Maf1* mutant worms [31]. However, the mechanisms underlying this longevity have not been directly examined.

To understand how the loss of Maf1 promotes longevity, we studied the role of Maf1 in aging, mainly in female fruit flies. We find that *Drosophila Maf1* (*dMaf1*) knockdown specifically in adult neurons extends lifespan. We present evidence that neuronal *dMaf1* knockdown acts by counteracting the loss of 5S rRNA expression with age and the resulting loss of protein synthetic capacity. This neuronal increase in translation promotes health both in wild type flies and in a model of C9Orf72-repeat neurotoxicity. Our study establishes a paradigm where enhancing translation capacity, specifically in neurons, promotes healthy aging.

## Results

### *dMaf1* knockdown extends lifespan but does not protect against obesity in flies

Previous work has demonstrated that loss of function in the *Maf1* gene can promote longevity in worms and mice [22,31]. We set out to investigate whether the effect of Maf1 on aging is evolutionarily conserved in the fruit fly *Drosophila melanogaster*. We chose to perform our experiments mainly in females, since female fruit flies show greater plasticity of lifespan (e.g., [32,33]) and the longevity due to Maf1 loss-of-function was observed in female but not male mice [22]. We used the RU486 inducible *GeneSwitch* system together with RNA interference (RNAi) to knock down *dMaf1* specifically in adulthood (unless otherwise noted, all knockdowns started from day two post-eclosion). We validated a *dMaf1^RNAi* line (V109142) by driving it with a constitutive, ubiquitous *Daughterless-gal4* (*Da-gal4*) driver: *dMaf1* mRNA levels were

halved in females (Fig 1A). There was a concomitant increase in pre-tRNA levels indicating that this level of *dMaf1* knockdown was sufficient to induce Pol III activity (Fig 1B). This increase was large and similar in magnitude to those observed by others [22,34]. Driving the expression of this RNAi line ubiquitously from day 2 post-eclosion with *Actin5c-GeneSwitch* (*Act5c-GS*) (Fig 1C) or *Daughterless-GeneSwitch* (*Da-GS*) (Fig 1D) was sufficient to significantly extended female lifespan, indicating *dMaf1* acts in adulthood to curtail longevity in the female fruit fly.

In mice, null mutation of *Maf1* confers resistance to diet-induced obesity [22]. To investigate whether it is the same case in *Drosophila*, we measured the levels of triacylglycerols (TAGs) in females fed an obesogenic, high-sugar diet [35], while ubiquitously inducing *dMaf1^RNAi* from adulthood. Lowering of *dMaf1* during adulthood did not counter the increased TAG levels observed after high-sugar feeding (Fig 1E and 1F). Furthermore, and consistent with the lack of effect on obesity, ubiquitous suppression of *dMaf1* was unable to combat the lifespan shortening effect of the high-sugar diet (S1A and S1B Fig), and indeed may even exacerbate it. Note, however, that the effect on obesity demonstrated in mice may also be observed in *Drosophila* with a more severe loss of *Maf1* function, e.g., resulting from a gene deletion. Overall, we found that the function of *Maf1* in aging was conserved in flies and could be uncoupled from an effect on obesity.

## Neuron-specific loss of *dMaf1* extends lifespan

Consistent with the longevity effects of reduced protein synthesis observed across species, we have shown that partial inhibition of Pol III extends chronological lifespan in budding yeast, as well as organismal lifespan in worms and female but not male flies [10,36]. In female fruit flies, the beneficial effects arise from specific cell types: attenuating the activities of Pol III in the gut, including its resident stem cells, promotes organismal longevity [10]. In contrast, loss of *dMaf1* increases Pol III activity. This prompted us to consider the possibility that *dMaf1* knockdown could be promoting longevity from a different organ. Publicly available mRNA single-cell expression data [37] indicate *dMaf1* is highly expressed in neurons, more highly than in other cell types, including the enterocytes, enteroendocrine cells, stem cell and enteroblasts in the gut (Fig 2A). Similarly, Maf1 expression is high in the mouse nervous system [28]. We also noted that age-related changes in Pol III activity were different between the brain and the gut. When the two organs were isolated from the same female flies, and the transcript levels detected were scaled to total DNA to avoid confounding effects of the substantial, known, age-related change in total cellular RNA content in the brain [38], we observed that the levels of *pre-tRNA^His*, *pre-tRNA^Ile*, *pre-tRNA^Leu* and *pre-tRNA^Tyr* all decreased in concert from day 7 to day 42 in the brain, indicating a reduction in Pol III activity (Fig 2B). In contrast, in the gut, we did not detect a concerted downregulation of these four pre-tRNA: although there was an age-related decrease in the levels of *pre-tRNA^His* and *pre-tRNA^Ile*, we observed a marked upregulation of *pre-tRNA^Leu* and a subtle increase in *pre-tRNA^Tyr*, indicating a tRNA-specific effect of aging on Pol III activity in that organ (Fig 2C). Similar results were obtained when *Actin5C* expression was used for normalization (S2A Fig). All this led us to consider whether suppression of *dMaf1* specifically in adult neurons would increase lifespan.

We induced *dMaf1^RNAi* in neurons of adult female flies using the *Elav-GS^Tricoire* driver, starting from day 2 of adulthood. This resulted in a reduction of *dMaf1* mRNA levels within the brain (~40% reduction, S2B Fig) and a significant lifespan extension (Fig 2D). This longevity phenotype was highly robust: we observed it in additional independent trials (S2C and S2D Fig), using two additional drivers (S2E and S2F Fig) as well as another, independent RNAi line (RNAi line validated in S2G and S2H Fig; S2I and S2J Fig). Interestingly, we found that neuronal *dMaf1* knockdown could also extend lifespan in male flies, albeit more modestly (S2K and S2L Fig; note that these are the only experiments we performed in males). Furthermore, late-onset (day 21), neuronal *dMaf1* knockdown was sufficient to extend female lifespan (Fig 2F). Longevity was not observed in driver- or RNAi-alone controls, indicating it is not an artifact of RU486 feeding (S2M, S2N, S2O, S2P, and S2Q Fig). Neuronal *dMaf1* knockdown also improved resistance to $H_2O_2$ (S2R Fig), a *Maf1* loss-of-function phenotype previously observed in *C. elegans* [31], with no such effect observed in the driver alone control (S2S Fig). Note that the longevity of the pan-neuronal *dMaf1* knockdown could not be recapitulated by driving *dMaf1^RNAi* in insulin-producing cells in the brain, whose role in aging is well documented [39] (S2T Fig). The reduction in neuronal *dMaf1* did not affect feeding

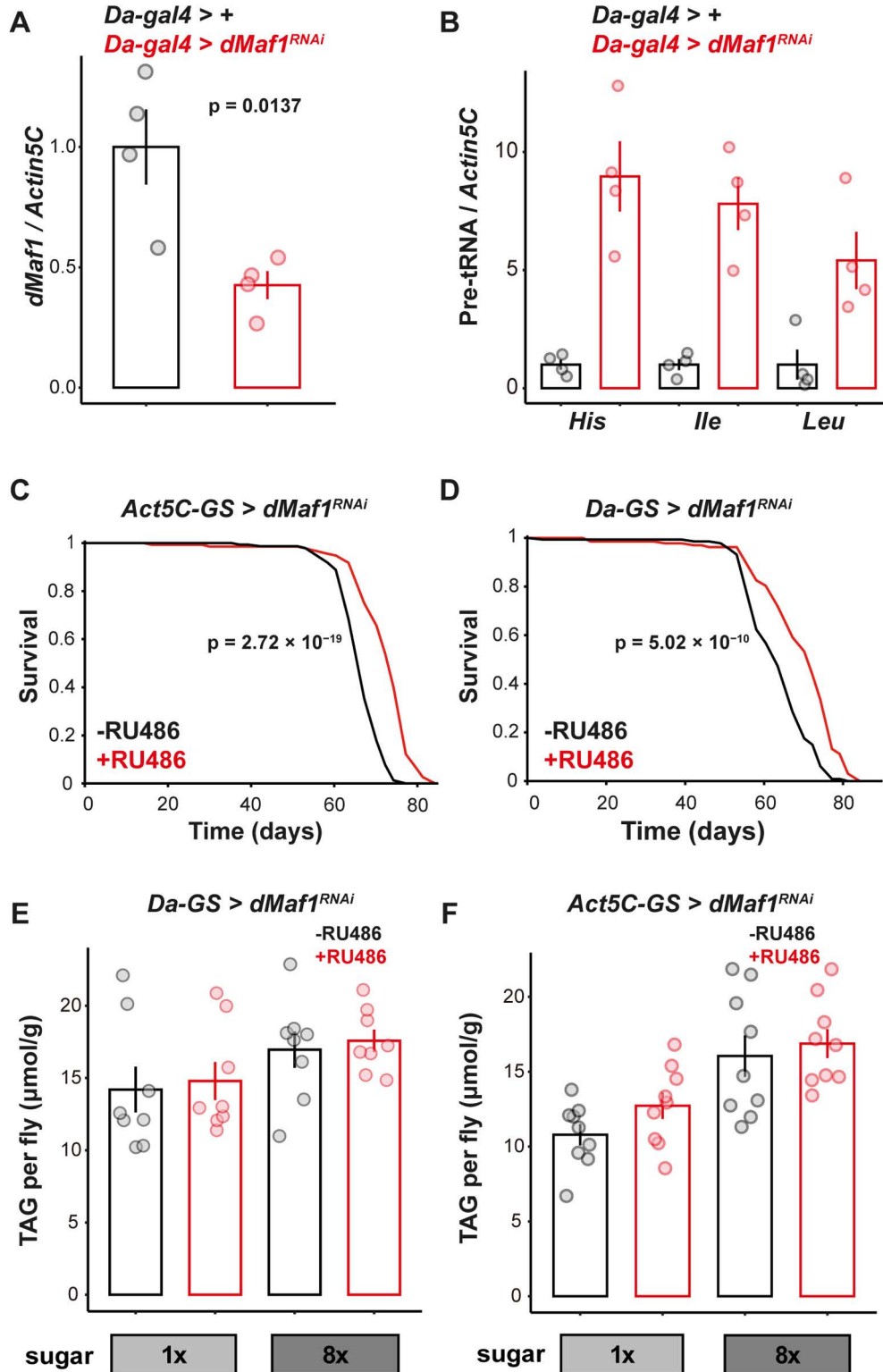

**Fig 1. Loss of *dMaf1* extends lifespan but may not protect against obesity. A,** qPCR quantification normalized to *Actin5C* of *dMaf1* mRNA ($n = 4$ biologically independent samples, $p = 0.0137$, *Student t test*), and **B,** pre-tRNAs [$n = 4$ biologically independent samples, genotype effect, $p < 1 \times 10^{-4}$, pre-tRNAs effect, $p = 0.1909$, genotype-by-pre-tRNAs interaction, $p = 0.1909$, Linear Model (*LM*)] in female flies expressing *dMaf1*[RNAi(V109142)] at day 14. **C,**

Lifespan of females with adult-specific, ubiquitous induction of $dMaf1^{RNAi(V109142)}$ driven by $Act5C\text{-}GS$ (−RU486: $n=122$ dead/26 censored flies, +RU486: $n=120/19$, $p=2.72\times10^{-19}$, $log\text{-}rank\ test$). **D**, Lifespan of females with adult-specific, ubiquitous induction of $dMaf1^{RNAi(V109142)}$ driven by $Da\text{-}GS$ (−RU486: $n=123/26$, +RU486: $n=114/35$, $p=5.02\times10^{-10}$, $log\text{-}rank\ test$). **E**, TAG levels of females at day 21 after feeding a high sugar diet or standard diet and combined with adult-specific, ubiquitous induction of $dMaf1^{RNAi(V109142)}$ with $Da\text{-}GS$ ($n=8$ biologically independent samples, sugar effect, $p=0.0371$, RU486 effect, $p=0.6370$, RU486-by-sugar interaction, $p=0.9916$, $LM$) or **F**, $Act5C\text{-}GS$ ($n=8$ biologically independent samples, sugar effect, $p<1\times10^{-4}$, RU486 effect, $p=0.1842$, RU486-by-sugar interaction, $p=0.5901$, $LM$). Bar charts indicate mean ± SEM with points showing individual values. For induction with GeneSwitch drivers, RU486 was supplied in the food from day 2 of adulthood. Data underlying the graphs in this figure can be found in S1 Data.

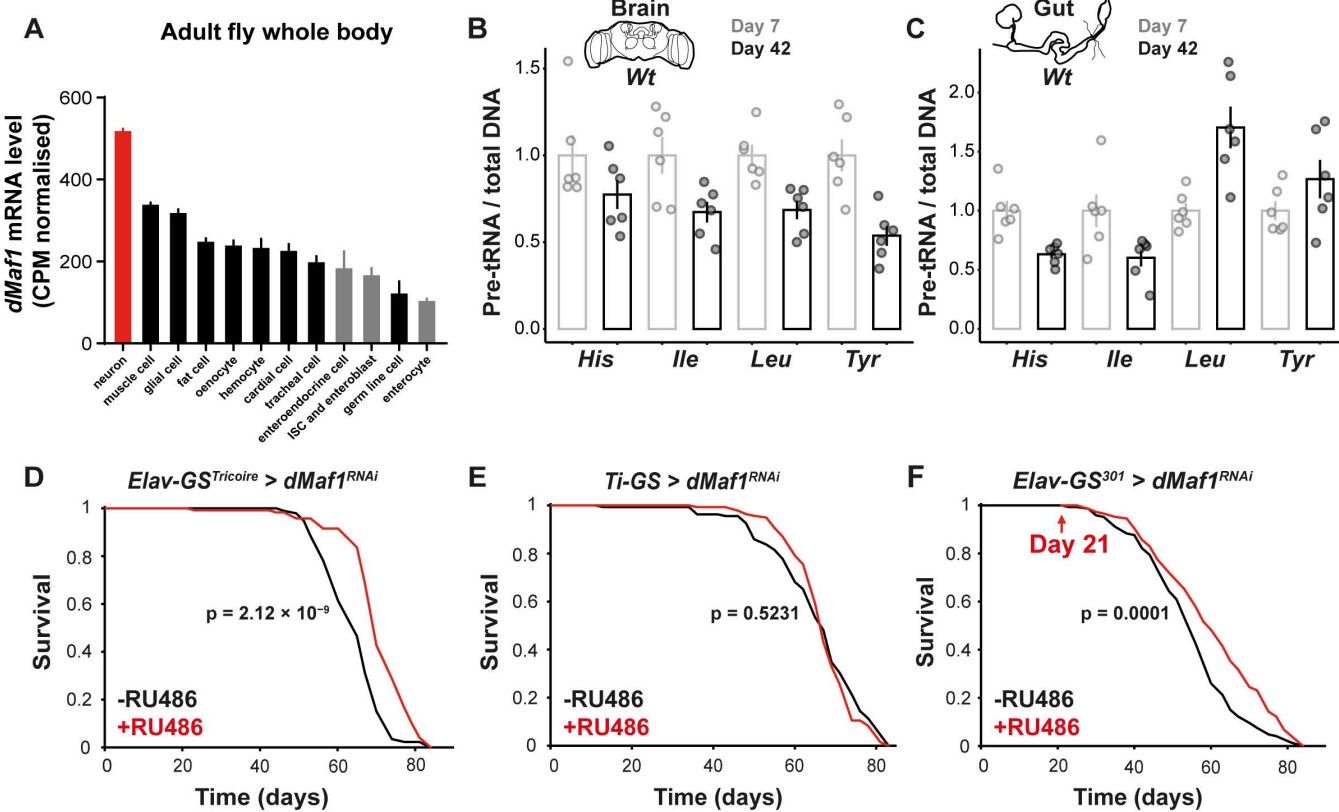

**Fig 2. Loss of *dMaf1* specifically in neurons extends lifespan. A**, *dMaf1* mRNA expression in adult cell types, as normalized counts per million (CPM) from single cell RNA-sequencing experiments in SCope shown for neurons (red) and other cell types (gut cells - gray, all others - black). **B**, qPCR quantification normalized by total DNA of pre-tRNAs in female brains [$n=6$ biologically independent samples, age effect, $p<1\times10^{-4}$, pre-tRNAs effect, $p=0.5499$, age-by-pre-tRNAs interaction, $p=0.5499$, *Linear Model* (*LM*)] or **C**, guts ($n=6$ biologically independent samples, age effect, $p=0.5194$, pre-tRNAs effect, $p<1\times10^{-4}$, age-by-pre-tRNAs interaction, $p<1\times10^{-4}$, *LM*) from same wild-type flies that were either 7 or 42 days old. **D**, Lifespan of females with adult-specific, pan-neuronal induction of $dMaf1^{RNAi(V109142)}$ driven by $Elav\text{-}GS^{Tricoire}$ (−RU486: $n=90$ dead/6 censored flies, +RU486: $n=112/7$, $p=2.12\times10^{-9}$, $log\text{-}rank\ test$). **E**, Lifespan of females with adult-specific, whole gut induction of $dMaf1^{RNAi(V109142)}$ driven by $Ti\text{-}GS$ (−RU486: $n=132/33$, +RU486: $n=137/9$, $p=0.5231$, $log\text{-}rank\ test$). **F**, Lifespan of females with later-life (from day 21), pan-neuronal induction of $dMaf1^{RNAi(V109142)}$ driven by $Elav\text{-}GS^{301}$ (−RU486: $n=146/0$, +RU486: $n=147/0$, $p=1\times10^{-4}$, $log\text{-}rank\ test$). For induction with GeneSwitch drivers, RU486 was supplied in the food from day 2 of adulthood except in F. Data underlying the graphs in this figure can be found in S1 Data.

behavior or egg laying (S2U and S2V Fig). The longevity effect appeared to arise specifically from neurons: knockdown of *dMaf1* in the female fat body (S2W Fig), or the gut (Figs 2E, S2X and S2Y) did not extend lifespan irrespective of the driver used, even though the increases in pre-tRNA levels in the gut and the fat body were similar to those observed in the brain upon *Maf1* knockdown (S2Z Fig). Still, it is possible that a different level of knockdown in those organs could provide some benefit. Note that the expression of neuronal drivers used in our study is restricted to neurons due to

well-characterized, neuron-specific expression of the *Elav* or *nSyb* gene promoters [40,41]; indeed we confirmed that *ElavGS301* was not active in glia, the other cell type in the *Drosophila* brain (S3A Fig). Additionally, published single cell RNA-Seq data indicate *Maf1* expression is low in glia [42] (S3B Fig). Thus, *dMaf1* appears to act specifically in neurons to promote aging.

## Loss of *dMaf1* within neurons delays age-related loss of organismal function

To further examine the effects of neuronal Maf*1* on aging, we first tested if adult-onset (from day 2), neuronal-specific *dMaf1* knockdown can guard against the age-related changes in the function of the neuromuscular system. We measured the ability of female flies to climb a vertical surface at different times in their life using a negative geotaxis (climbing) assay and analyzed the data using a *linear model* (*LM*). Adult-onset, neuronal *dMaf1* knockdown delayed the age-related decline in climbing ability in females (Fig 3A). The statistical significance of this delay was confirmed as a significant age-by-RU486 interaction in the *LM* ($p = 0.0046$). Indeed, even a later-onset knockdown (from day 21) appeared sufficient to counteract the loss in climbing ability (S4A Fig). RU486 treatment started either early or late did not affect the climbing ability of the driver-alone control (S4B and S4C Fig).

During aging, *Drosophila*'s sleep fragments: the length of sleep decreases and the number of sleep bouts increases [43]. Next, we monitored activity and sleep patterns in young and old females over 24 h (one 12 h:12 h day:night cycle). Neuronal *dMaf1* knockdown did not affect day or night activity, day sleep or day sleep bouts (S4D, S4E, S4F, and S4G Fig), but it prevented the decline in the length of total night sleep and the increase in night sleep bouts during aging, indicating greater sleep consolidation (Fig 3B; age-by-RU486 interaction $p = 0.0101$ and 3C; age-by-RU486 interaction $p = 0.0102$). There was no effect of RU486 on these sleep phenotypes in the driver alone control (S4H and S4I Fig).

Changes in one organ can affect the aging of others [11,44]. The gut of older flies loses its ability to act as a barrier and this phenotype can be easily scored with a "smurf" assay where gut leakage is assessed with a blue food dye. Neuron-specific *dMaf1* knockdown reduced the loss of gut barrier function during aging in females (Fig 3D). No such effect was observed in the driver-alone control (S4J Fig). The beneficial effects on both neuromuscular performance and gut health of the neuronal *dMaf1* knockdown were confirmed with an independent RNAi line (S4K Fig; age-by-RU486 interaction $p = 0.0413$ and S4L Fig). Overall, our analyses show that a reduction in neuronal *dMaf1* is sufficient to improve the health of multiple organ systems, including distal organs such as the gut.

## Reduction in neuronal *Maf1* requires Pol III initiation on the *5S* rRNA to extend lifespan

Maf1 is a global Pol III repressor, but in certain contexts it has been shown to regulate Pol II-transcribed genes [15,20,24]. To examine if *dMaf1* acts via Pol III to impact aging in female fruit flies, we co-induced a validated RNAi line against the largest Pol III subunit *Polr3A* (previously named *dC160*, see [45]) together with *dMaf1* RNAi in neurons. We found that *Polr3A* knockdown abolished the lifespan extension of a *dMaf1* knockdown (Fig 4A), and *Polr3A* knockdown alone did not affect lifespan (S5A Fig). We confirmed a significant difference in the response to the RU486 inducer in the presence or absence of the *Polr3A* knockdown using *Cox Proportional Hazards* (*CPH*) analysis (RU486-by-genotype interaction, $p = 1 \times 10^{-4}$). Hence, *dMaf1* acts through Pol III for longevity.

Pol III transcribes a number of different genes with different cellular functions [17]. To gain a better understanding of how neuronal *dMaf1* impacts female lifespan, we used RNAi lines targeting transcription factors (TFs) required for Pol III initiation, namely TFIIIA, TFIIIC, and SNAPc, as these define specific subsets of Pol III transcribed genes. TFIIIA is required for transcription of the 5S rRNA, TFIIIC for both 5S and tRNAs, and SNAPc for other RNAs such as the U6 [17,46]. The knockdown of either *GTF3A* (*CGCG9609*, the *Drosophila* orthologue of the human *GTF3A* gene encoding TFIIIA), *GTF3C3* (*CG8950*, the *Drosophila* orthologue of the human *GTF3C3* gene encoding a subunit of TFIIIC) or *Pbp95* (a SNAPc subunit) in adult neurons alone did not affect fly lifespan (S5B, S5C, and S5D Fig). Knockdown of either

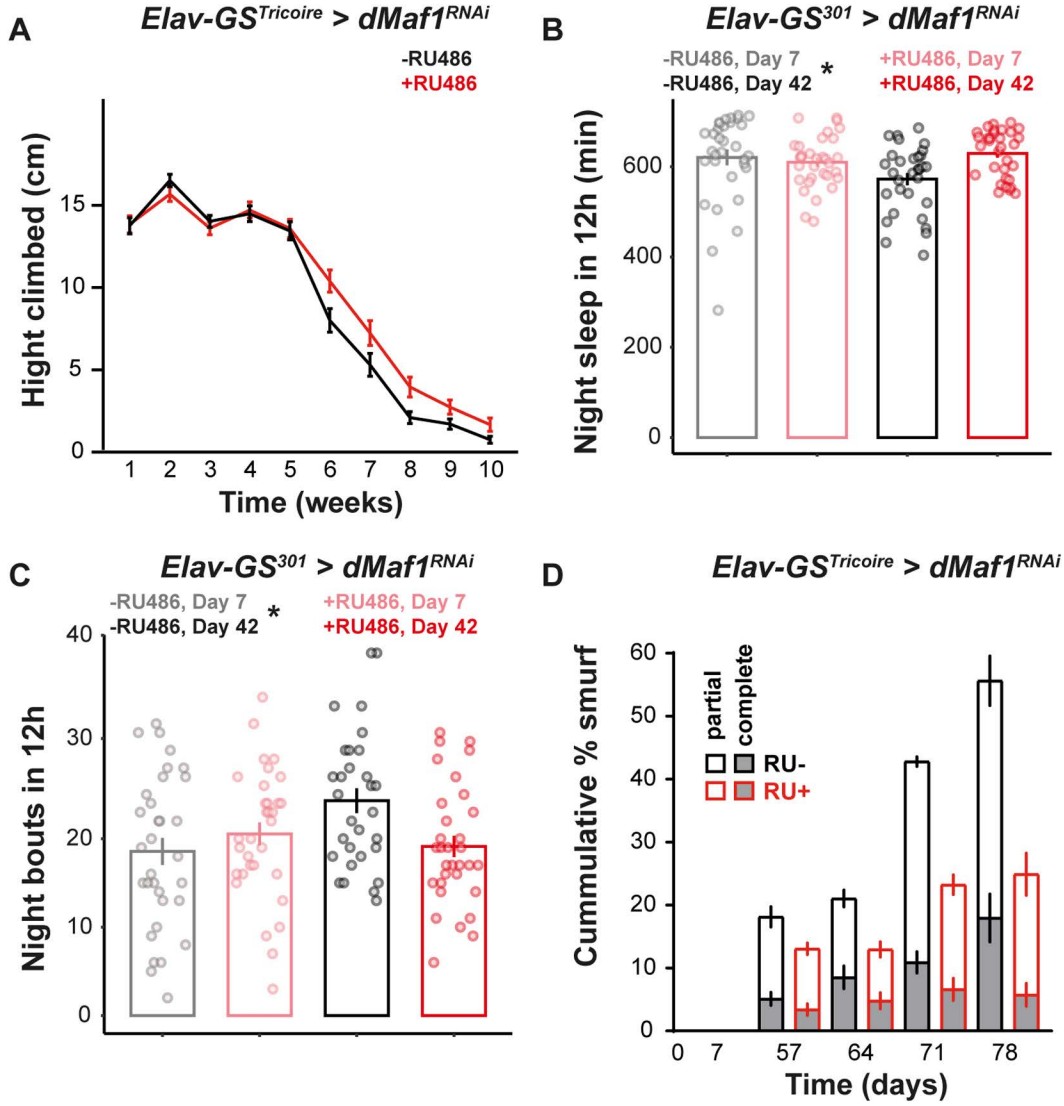

**Fig 3. Loss of *dMaf1* in neurons improves old-age health. A**, Height climbed during negative geotaxis assays by female flies with adult-specific, pan-neuronal induction of *dMaf1*<sup>RNAi(V109142)</sup> driven by *Elav-GS*<sup>Tricoire</sup> [$n = 44–70$ flies, RU486 effect, $p = 0.0081$, age effect, $p < 1 × 10^{-4}$, RU486-by-age interaction, $p = 0.0046$, *Linear Model* (*LM*)]. **B**, Quantification of females' night sleep ($n = 32$ individual flies, age effect, $p = 0.2739$, RU486 effect, $p = 0.0799$, RU486-by-age interaction, $p = 0.0101$, *LM*) and **C**, night bouts ($n = 32$ individual flies, age effect, $p = 0.1160$, RU486 effect, $p = 0.2604$, RU486-by-age interaction, $p = 0.0102$, *LM*) of 7- and 42-day old females with adult-specific, pan-neuronal induction of *dMaf1*<sup>RNAi(V109142)</sup> with the *Elav-GS*<sup>301</sup> driver. **D**, Cumulative proportion of partial and complete smurfs in female flies with adult-specific, pan-neuronal induction of *dMaf1*<sup>RNAi(V109142)</sup> driven by *Elav-GS*<sup>Tricoire</sup> ($n = 92–380$ flies, RU486 effect, $p = 0.0160$, age effect, $p < 1 × 10^{-4}$, RU486-by-age interaction, $p = 0.3833$, *ordinal logistic regression*). For induction with GeneSwitch drivers, RU486 was supplied in the food from day 2 of adulthood. Data underlying the graphs in this figure can be found in S1 Data.

*GTF3C3* or *GTF3A* abolished the lifespan extension resulting from *dMaf1* knockdown (Fig 4B and 4C), whereas loss of *Pbp95* did not (Fig 4D). Note that all the RNAi lines used resulted in a significant knockdown of the target transcript and the expected pre-adult lethality when combined with a ubiquitous, constitutive driver (*Da-gal4*, S5E–S5I Fig); knockdown of TFIIIA reduced 5S levels (S5J Fig). In summary, the genetic analysis revealed TFIIIA and TFIIIC as essential for the longevity benefit of *dMaf1* knockdown.

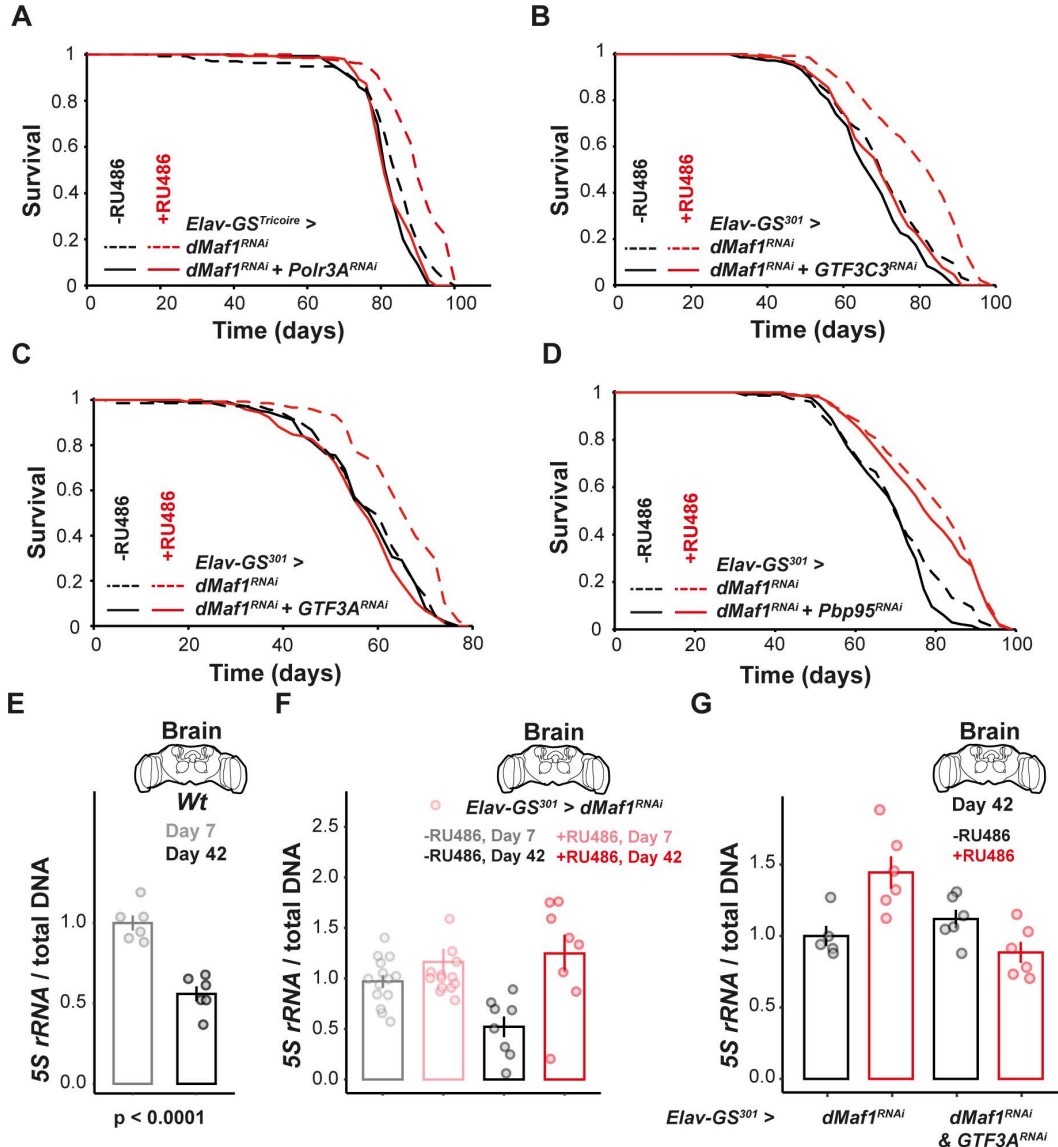

**Fig 4. Loss of neuronal d*Maf1* extends lifespan by maintaining 5S rRNA expression during aging. A**, Lifespan of females with adult-specific, pan-neuronal induction of *dMaf1*$^{RNAi(V109142)}$ with or without co-induction of *Polr3A*$^{RNAi}$, driven by *Elav-GS*$^{Tricoire}$ [*dMaf1*$^{RNAi(V109142)}$ alone, −RU486: *n* = 134 dead/2 censored flies, +RU486: *n* = 131/3, *p* = 1.18 × 10$^{-10}$, *log-rank test*; *dMaf1*$^{RNAi(V109142)}$ with *Polr3A*$^{RNAi}$, −RU486: *n* = 149/1, +RU486: *n* = 145/5, *p* = 0.3164, *log-rank test*; genotype effect, *p* = 8.62 × 10$^{-5}$, RU486 effect, *p* = 3.35 × 10$^{-10}$, genotype-by-RU486 interaction, *p* = 1.02 × 10$^{-4}$, *Cox Proportional Hazards* (*CPH*)]. **B**, Lifespan of females with adult-specific, pan-neuronal induction of *dMaf1*$^{RNAi(V109142)}$ with or without co-induction of *GTF3C3*$^{RNAi}$, driven by *Elav-GS*$^{301}$ (*dMaf1*$^{RNAi(V109142)}$ alone, −RU486: *n* = 151/1, +RU486: *n* = 145/4, *p* = 2.41 × 10$^{-9}$, *log-rank test*; *dMaf1*$^{RNAi(V109142)}$ with *GTF3C3*$^{RNAi}$, −RU486: *n* = 143/4, +RU486: *n* = 145/7, *p* = 0.0371, *log-rank test*; genotype effect, *p* = 6.50 × 10$^{-4}$, RU486 effect, *p* = 9.78 × 10$^{-10}$, genotype-by-RU486 interaction, *p* = 0.0029, *CPH*). **C**, Lifespan of females with adult-specific, pan-neuronal induction of *dMaf1*$^{RNAi(V109142)}$ with or without co-induction of *GTF3A*$^{RNAi}$, driven by *Elav-GS*$^{301}$ (*dMaf1*$^{RNAi(V109142)}$ alone, −RU486: *n* = 143/0, +RU486: *n* = 140/7, *p* = 4.23 × 10$^{-9}$, *log-rank test*; *dMaf1*$^{RNAi(V109142)}$ with *GTF3A*$^{RNAi}$, −RU486: *n* = 144/5, +RU486: *n* = 138/6, *p* = 0.2169, *log-rank test*; genotype effect, *p* = 0.7240, RU486 effect, *p* = 2.84 × 10$^{-8}$, genotype-by-RU486 interaction, *p* = 1.86 × 10$^{-6}$, *CPH*). **D**, Lifespan of females with adult-specific, pan-neuronal induction of *dMaf1*$^{RNAi(V109142)}$ with or without conduction of *Pbp95*$^{RNAi}$, driven by *Elav-GS*$^{301}$ (*dMaf1*$^{RNAi(V109142)}$ alone, −RU486: *n* = 151/1, +RU486: *n* = 145/4, *p* = 2.41 × 10$^{-9}$, *log-rank test*; *dMaf1*$^{RNAi(V109142)}$ with *Pbp95*$^{RNAi}$, −RU486: *n* = 135/0, +RU486: *n* = 142/10, *p* = 7.42 × 10$^{-13}$, *log-rank test*; genotype effect, *p* = 0.0050, RU486 effect, *p* = 8.65 × 10$^{-10}$, genotype-by-RU486 interaction, *p* = 0.1710 *CPH*). **E**, qPCR quantification of 5S rRNA in female brains of wild-type flies (*n* = 6 biologically independent samples, *p* < 1 × 10$^{-4}$, *Student t test*) or **F**, those with adult-specific, pan-neuronal induction of *dMaf1*$^{RNAi(V109142)}$ driven by *Elav-GS*$^{301}$, relative to total DNA, at 7 and 42 days. [*n* = 14 and 8 biologically independent samples respectively, age effect, *p* = 0.1503, RU486 effect, *p* = 0.0026, RU486-by-age interaction, *p* = 0.0385, *Linear Model* (*LM*)]. G, qPCR quantification of 5S rRNA in female brains at 42 days after *dMaf1*$^{RNAi(V109142)}$ was induced alone or with *GTF3A*$^{RNAi}$ by *Elav-GS*$^{301}$, relative to total DNA (*n* = 5−6 biologically independent samples, genotype effect *p* = 0.017, RU486 effect *p* = 0.22, genotype-by-RU486 interaction *p* = 7 x 10$^{-4}$, *LM*). For induction with GeneSwitch drivers, RU486 was supplied in the food from day 2 of adulthood. Data underlying the graphs in this figure can be found in S1 Data.

As both TFIIIC and TFIIIA are required for the transcription of *5S* rRNA, we focused our attention on this rRNA and examined what happens to its expression with age in female flies. To avoid the potential confounding effects of the age-related reduction in neuronal RNA concentration previously observed [38], we normalized *5S* rRNA expression to total DNA extracted from the same samples. We found that 5S rRNA levels were significantly reduced during aging in wild-type, female brains (Figs 4E and S5K). Interestingly, 5S rRNA expression was not changed in the gut during aging (S5L and S5M Fig), consistent with the differential effect of aging on Pol III activity in the two organs. In the brain, we found that *Maf1* mRNA levels were not altered by age, but the levels of two Pol III subunits, *Polr3A* and *Polr3D*, were reduced, indicating that the loss of 5S expression may have been due to lower Pol III expression (S5N and S5O Fig). In concordance with lifespan results of the genetic epistasis analysis, loss of neuronal *dMaf1* resulted in slightly increased *5S* rRNA expression in young flies and provided substantial protection against the age-related decline in *5S* levels (Fig 4F, age-by-RU486 interaction $p=0.0385$). Similar results were observed when *5S* expression was normalized to *Act5C* mRNA (S5P Fig). As expected, induction of *GTFIIIA^{RNAi}* in neurons substantially reduced *5S* levels (S5Q Fig) and prevented the *Maf1* knockdown from rescuing the expression of *5S* rRNA in older flies (Fig 4G, genotype-by-RU486 interaction $p=7 \times 10^{-4}$; see S5R Fig for *5S* relative to *Actin5C*). Overall, our findings are consistent with neuronal *dMaf1* knockdown increasing lifespan by countering an age-related loss in *5S* expression though Pol III activation.

## Loss of *dMaf1* protects against a decline in protein synthesis and C9orf72 repeat toxicity

The *5S* rRNA is an essential constituent of the ribosome [47]. Our observations prompted us to further examine the role of the neuronal protein synthetic capacity in the longevity attained by neuronal *dMaf1* knockdown. We monitored the rates of protein synthesis in isolated female fly brains using puromycin incorporation; we chose to assess protein synthesis in the whole brain rather than isolated neurons to ensure robustness and because ~90% of cells in the fly brain are neurons [38]. Consistent with extensive previous work in a number of species [4], we found that protein synthesis in female *Drosophila* brains was significantly reduced with age (Figs 5A and S6A). Loss of neuronal *dMaf1* slightly increased protein synthesis rates in young flies and, importantly, significantly rescued the decline in protein synthesis in older brains; this rescue was observed both when we assayed young and old flies at the same time (Figs 5B and S6B) as well as when we followed a cohort of flies though time (S7A–S7C Fig). Note that this rescue of protein synthesis appeared less pronounced than the rescue of 5S expression, indicating that additional factors or processes required for protein synthesis are compromised with age. To validate the physiological relevance of this increase in translation in older flies, we combined the neuronal *dMaf1* knockdown with RNAi against *RpS5a* encoding an essential ribosomal protein. This loss of function in a ribosomal protein was sufficient to abolish the longevity effects of neuronal *dMaf1* knockdown (Fig 5C), while *RpS5a* knockdown alone did not affect lifespan (S7D Fig). Overall, our data indicate that attenuated *dMaf1* function in neurons extends female lifespan by preserving neuronal protein synthesis against the effects of aging. This increase in protein synthesis could also increase protein turnover, and indeed autophagy is induced in *Maf1* knockout mice [22]. Consistent with this, we observed an increase in lysotracker staining in brains upon *Maf1* knockdown (S7E Fig), indicating an increased lysosomal compartment, which suggests an increase in autophagy in *Drosophila* females as in mice. However, further work is required to directly demonstrate an increase in autophagic flux.

Amyotrophic lateral sclerosis (ALS) and frontotemporal dementia (FTD) are neurodegenerative diseases whose most common genetic cause is a hexanucleotide repeat expansion within the *C9orf72* gene [48]. The *C9orf72* repeat RNA is translated into toxic dipeptide repeats that can interact with ribosomal proteins and inhibit translation [49]; this inhibition of translation is an important mechanism of toxicity. Since neuronal *dMaf1* knockdown could rescue the loss in translation capacity in aged but otherwise healthy female flies, we examined its ability to reduce the toxicity of the *C9orf72* repeat RNA in *Drosophila*. Indeed, loss of *dMaf1* also partially rescued the short lifespan of flies expressing the expanded 36 GGGGCC (G4C2) repeats (36R) in adult neurons (Fig 5D). Interestingly, we found that the induction of the repeats caused an increase in Maf1 levels in the fly head, potentially mediating toxicity of 36 GGGGCC (S7F Fig). Hence, loss

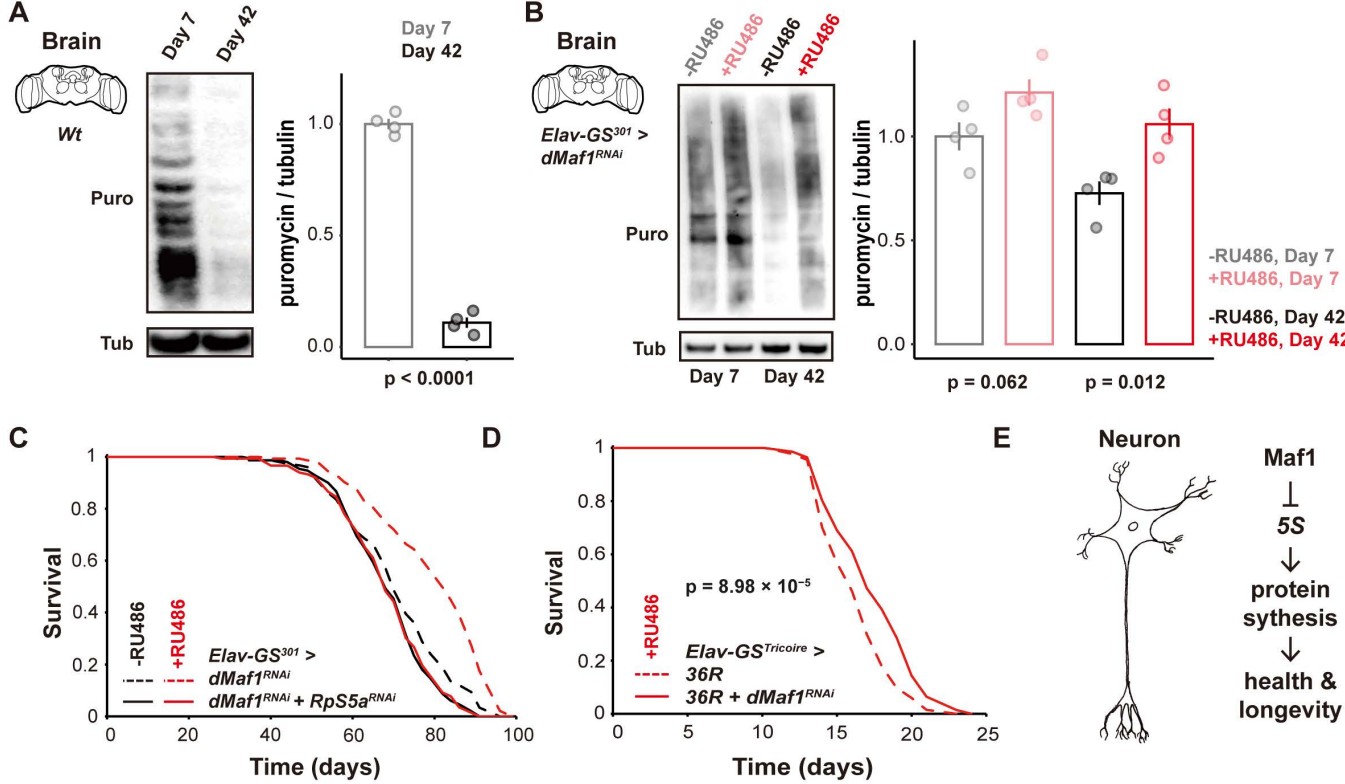

**Fig 5. Loss of neuronal *dMaf1* extends lifespan through increased protein synthesis and partially rescues the toxicity of the C9Orf72 repeat RNA. A**, Representative western blot of puromycin incorporation in female brains of wild-type flies at 7 and 42 days of age ($n=4$ biologically independent samples, $p<1\times10^{-4}$, *Student t test*). **B**, Representative western blots and quantification of puromycin incorporation in brains of female flies with adult-specific, pan-neuronal induction of *dMaf1^RNAi(V109142)* driven by *Elav-GS^301* at 7 and 42 days of age ($n=4$ biologically independent samples; −RU486 vs. +RU486 7 days $p=0.062$, 42 days $p=0.012$, *Student t test*; effect of age $p=7.2\times10^{-3}$, effect of RU486 $p=1.4\times10^{-3}$, RU486-by-age interaction $p=0.37$, *LM*). The flies of the two different ages were assayed at the same time. **C**, Lifespan of females with adult-specific, pan-neuronal induction of *dMaf1^RNAi(V109142)* with or without conduction of *RpS5a^RNAi*, driven by *Elav-GS^301* [*dMaf1^RNAi(V109142)* alone, −RU486: $n=151$ dead/1 censored flies, +RU486: $n=143/4$, $p=2.05\times10^{-9}$, *log-rank test*; *dMaf1^RNAi(V109142)* with *RpS5a^RNAi*, −RU486: $n=150/1$, +RU486: $n=147/2$, $p=0.9793$, *log-rank test*; genotype effect, $p=0.0033$, RU486 effect, $p=6.17\times10^{-10}$, genotype-by-RU486 interaction, $p=8.74\times10^{-6}$, *Cox Proportional Hazards* (*CPH*)]. **D**, Lifespan of females with adult-specific, pan-neuronal induction of *dMaf1^RNAi(V109142)* with *36R* driven by *Elav-GS^Tricoire* (*36R* alone: $n=134/5$, *36R* with *dMaf1^RNAi(V109142)*: $n=141/1$, $p=8.98\times10^{-5}$, *log-rank test*). **E**, Model of how Maf1 activity in neurons impacts aging. For induction with GeneSwitch drivers, RU486 was supplied in the food from day 2 of adulthood. Data underlying the graphs in this figure can be found in S1 Data.

of *dMaf1* within neurons promotes neuronal protein synthetic capacity to improve organismal health during both normal aging and in a disease context.

## Discussion

Protein synthesis underlies many fundamental neuronal functions: for example, it is required for long-term memory formation [3] and plays a role in neural plasticity [7]; it is also elaborately controlled, with translation of specific mRNAs localized to subcellular regions such as axons [6,7]. During aging, the capacity to synthesize proteins declines in many tissues and organs, including the brain [2–4]. This decline is conserved across different animals, such as flies and mammals [4], and may be due, at least in part, to an inappropriate or insufficient provision of ribosomal components (e.g., [50,51]). Our work shows that counteracting this decline in neurons of the adult female fruit fly, specifically by reducing the activity of the Pol III inhibitor Maf1 and consequently increasing the Pol III transcription, can be beneficial for old-age health and

longevity of the entire organism. The effects on the overall organismal health may be both direct, e.g., preventing a life- or health-limiting neuronal pathology such as caused by *C9orf72* repeat RNA, and indirect, e.g., hormonal or metabolic. Our data pinpoint the neuronal *5S* rRNA as a key player in this mechanism (Fig 5E), but the potential contribution from other Pol III-transcribed RNAs, such as tRNAs, cannot be excluded. Interestingly, reduced translation has been implicated in pathology of neurodegenerative disease [49,52]. Our work supports the notion that this reduction in neuronal translation capacity is a link between physiological aging and disease.

Ubiquitous loss of function in *Maf1* also extends mouse lifespan [22]. We consider that it is likely that mammalian *Maf1* is also acting from neurons to impact aging of the whole animal. Indeed, a role for Maf1 in neuronal physiology has recently been described [27–30]. In this context, it is interesting that *Maf1* cell-intrinsically limits neuronal repair, so that a loss-of-function in *Maf1* significantly enhances neuronal plasticity and functional recovery after injury, probably through increased transcription of the components of the protein synthetic machinery [30]. Reduced *Maf1* function may act simply by allowing cells to synthetize more proteins required for appropriate cellular function. For example, it has been suggested that the decline in neuronal protein synthesis contributes to age-related memory impairments, due to the requirement for de novo protein synthesis in memory formation [3]. Indeed, high hippocampal *Maf1* levels negatively impact mouse learning and memory [28]. Additionally, increasing protein synthetic capacity may affect protein turnover rates, which have been documented to decline in the aging brain [53]: increasing synthesis may accelerate protein turnover, allowing clearance of damaged or misfolded proteins, thus, improving proteostasis, a key determinant of neuronal health [54]. Indeed, autophagy is induced in *Maf1* knockout mice [22] and appears similarly impacted by a reduction in *Maf1* in flies. Additionally, Maf1 knockdown may impact translational fidelity, which can have positive effects on fruit fly lifespan [14]. Overall, we propose that reduced function of *Maf1* is likely to boost health of mammalian neurons during aging, through the same or similar mechanisms as we describe here for the fruit fly.

Maf1 loss-function has been reported to extend female but not male lifespan [22]. Sexual dimorphism in the effects of longevity interventions is often observed in fruit flies and other animals, but we are only now starting to understand its molecular and physiological bases [32,33]. In this study, we focused on female flies and only examined the lifespan effect of neuronal *Maf1* knockdown in males. The lifespan of the males was extended, albeit more modestly than in females; however, a more pronounced sexual dimorphism may be revealed in other traits by subsequent studies, allowing for further investigation. Such subsequent investigation will allow for the important understanding of the root of the likely sexually dimorphic effects of Maf1 in mammals.

It is interesting to note that the increase in protein synthesis by inhibition of *Maf1* and consequent Pol III activation appears beneficial in neurons while the opposite is true in the fly gut, where the inhibition of Pol III in the resident stem cells promotes organismal longevity [10]. Curiously, the converse interventions, knockdown of *Maf1* in the gut or of a Pol III subunit in the brain, are not detrimental. This is likely due to other, unrelated and parallel processes that also drive aging. It can also explain why the ubiquitously implemented interventions, e.g., ubiquitous knockdown of *Maf1* or the heterozygous *Polr3D* mutant [10], show a lifespan extension phenotype. The differing requirements for Pol III activity in neurons and intestinal stem cells during aging also parallel the different effect age has on expression of Pol III-transcribed genes in the two cell type we describe in this study.

Furthermore, while the two interventions are reciprocal, the increased neuronal protein synthetic capacity and attenuated translation in intestinal stem cells may not be acting on aging by opposing mechanisms. Indeed, there is increasing evidence that a reduction in translation acts by triggering a stress response or by modulating gene expression, rather than by simply reducing bulk translation [55–59]. For example, impairments in protein translation result in the activation of Gcn4 and Atf-4 TFs, in yeast and worms, respectively. These factors, in turn, activate pro-longevity transcriptional programmes that enhance stress resistance and improve proteostasis [55,58]. Indeed, attenuation of Pol III activity induces a proteostatic response and an increased tolerance of a proteostatic challenge in worms, flies, and mice, where the beneficial effects appear to derive from changes in tRNA expression [60]. As an additional example of mechanism, attenuation

of translation early in fruit fly adulthood appears to provide organism-wide benefits by altering intercellular communication: during *Drosophila* adulthood, protein synthesis rates peak in early youth and drive the subsequent, later-age decline in health via hormonal signals that impact proteostasis [5].

Loss of proteostasis is a fundamental feature of aging and directly enhancing proteostasis extends organismal lifespan [54,61,62]. Decelerating translation may also, for example, enhance co-translational proteostasis, a process that is compromised with age [63]. This is consistent with recent suggestions that the observed reduction in translation later in fruit fly adulthood may be "adaptive" [5], at least in some cell types. Interestingly, attenuation of translation needs to be implemented in early adulthood to prevent the burst in protein translation that occurs in *Drosophila* youth to result in subsequent health benefits later in life [5], whereas neuronal *Maf1* knockdown and the consequent increase in neuronal translation are effective even at older ages, indicating that the two interventions may act in a temporally distinct manner. Again, they may both impact proteostasis as increased translation may in effect increase protein turnover, as discussed above. Indeed, they may represent two mechanisms to ensure proteostasis, one important for non-dividing and other for dividing cells.

Our ongoing work on Pol III in flies highlights how different interventions may benefit different cells: reduced Pol III activity is beneficial in the proliferative stem cells in the fly gut, while enhanced Pol III activity is beneficial in the non-proliferative neurons. Importantly, complementing a wealth of studies showing the benefits of attenuated protein synthesis for aging, this study uncovers a cell-type-specific benefit of increased translation capacity that is likely to be relevant in the context of mammalian, including human, aging.

## Materials and methods

### Fly husbandry

Fly stocks and experiments were maintained on a 12 h:12 h light/dark cycle at 25°C with 60% humidity, on the standard sugar/yeast/agar (SYA) medium containing 10% (w/v) brewer's yeast, 5% (w/v) sucrose and 1.5% (w/v) agar [64]. The outbred Dahomey background wild-type stock was obtained in Dahomey (now Benin) in 1970 and has been maintained in large population cages. The *white Dahomey* ($w^{Dah}$) or *vermillion Dahomey* ($v^{Dah}$) stocks were derived by incorporating the $w^{1118}$ or $v^1$ mutation, respectively, into the Dahomey background by extensive backcrossing. All fly stocks in this study were backcrossed for six or more generations into the outbred $w^{Dah}$ or $v^{Dah}$ background. Fly stocks used were *Actin5c-GS*, *Da-GS*, *Elav-GS^{Tricoire}*, *Elav-GS^{301}*, *nSyb-GS*, *Dilp2-GS*, *Ti-GS*, *5966-GS*, *5961-GS*, *S_1106-GS*, *Da-Gal4* (Bloomington #55850), *UAS-dMaf1^{RNAi}* (VDRC #109142), *UAS-dMaf1^{RNAi}* (Bloomington #64603), *UAS-nls-GFP* (a gift from Bruce Edgar), *UAS-Polr3A^{RNAi}* (VDRC #30512), *UAS-GTF3C3^{RNAi}* (Bloomington #58079), *UAS- GTF3A^{RNAi}* (Bloomington #34030), *UAS-Pbp95^{RNAi}* (Bloomington #67281), *UAS-RpS5a^{RNAi}* (VDRC #101472), *UAS-36R*.

### Lifespan assays

Lifespan assays were performed as described previously in detail [65]. In brief, crosses were set up in cages containing grape juice agar and live yeast, and eggs were collected within a 24-h window. Emerged adult flies were allowed to mate for 48 h and were subsequently lightly anesthetised with $CO_2$ and separated by sex to a density of 15 per vial. Females were used unless otherwise noted. Experimental vials contained either SYA food with 200 μM RU486 (Sigma, M8046) or the control food containing equivalent volume of the ethanol vehicle. Flies were transferred to fresh food three times a week with deaths/censors recorded. Vials were kept in DrosoFlippers (https://www.drosoflipper.com/). For later life induction, RU486 food was supplied from day 21. Data were collected and analyzed by *log-rank test* in Excel (template available at http://piperlab.org/resources/). Details of statistical analyses and number of flies per condition (*n*) are provided in figure legends. CPH analysis was performed using the *survival* package [66,67] in R [68].

## Triacylglycerol (TAG) and weigh measurements

A single female fly was weighed and then homogenized with 400 µl of 0.05% Tween-20 and TAG measured with Infinity TAG Reagent (Thermo Scientific). The highest/lowest measurements for each condition were removed to protect against potential outliers. Data were analyzed with JMP.

## Immunostaining

Female brains were dissected from 7-day-old flies in ice-cold PBS and immediately fixed in 4% formaldehyde for 30 min. Brains were washed 3 × 20 min in PBS-T (0.5% Tween20) and blocked for 1 h in 5% horse serum (Gibco) in PBS-T. Brains were then incubated overnight at 4°C in 1:200 anti-REPO (8D12, DSHB) in blocking solution, followed by 1hr of PBS-T washes and a further overnight incubation at 4°C in 1:250 goat-anti-mouse Alexa 568 (A11019, Thermo Fisher). Brains were then mounted in VECTASHIELD mounting medium with DAPI and imaged at 63× on a Zeiss 880 confocal microscope.

## Negative geotaxis (climbing) assays

Flies in five vials containing control food and five vials of food containing RU486 (200 µM) were used for climbing assay. At indicated times, flies were transferred to empty vials in DrosoFlippers to allow climbing of two vial heights. After acclimatizing for 30 min, flies were tapped to the bottom of the vials and allowed to climb upwards for 15 s (the time it takes young wild-type flies to nearly reach maximum height) while being video recorded. The final still images were analyzed in Fiji86 to collect the height climbed by individual fly. If the height could not be determined, the fly was not used in the analysis. Data were analyzed with a LM using JMP. The same cohort was continuously assayed.

## Activity and sleep analysis

Twenty-four-hour activity and sleep were monitored using the Drosophila Activity Monitor (DAM2) system and DAMSystem3 data acquisition software (Trikinetics). At indicated times, individual female flies form indicated treatment groups were placed in tubes with control food or food containing RU486 (200 µM). Flies were loaded into DAMs and placed into 25°C, 65% humidity incubator (Percival) with 12h:12h light:dark cycle. After one day acclimatization, data were obtained from a 24-h period (beginning at the onset of lights-on). A custom Microsoft Excel workbook [69] was used to calculate total activity counts per fly in the day and night periods, and total sleep minutes and number of sleep bouts per fly during day and night periods. Data from flies that appear dead were not used, and one outlier was removed from day or night activity and bouts after visual inspection. Data were analyzed with a LM using JMP.

## Smurf assays

For Smurf assay, the flies of indicated age were placed on SYA food with 2.5% (w/v) blue dye (FD&C blue dye no. 1, Fastcolors) for 48 h. Flies were scored as full smurfs if completely blue or partial smurfs if the dye had leaked out of the gut but not reached the head. The same cohort was continuously assayed. Data were analyzed with ordinal logistic regression using JMP.

## Fecundity assay

Six to ten vials with the density of 15 7-day old females kept on control food and food containing RU486 (200 µM) were used to count the eggs laid over a 24-h period. The number of eggs laid per fly per day was calculated. Data were analyzed with LM using JMP.

## Capillary Feeder (CAFE) assay

A single fly was added into a 7 mL bijou vial containing 1 mL 1% agar (w/v) sealed with Parafilm (Alpha Laboratories, Hampshire, UK). Holes were made on Parafilm for adequate air circulation, and a 5 ml disposable glass capillary tube

(Camag, Muttenz, Switzerland) containing liquid food (5% sucrose (w/v), 2% BD Bacto yeast extract (w/v)) supplemented with 0.5 mg/mL FD&C blue dye no. 1 (Fastcolors) to aid measurement of feeding was provided through the Parafilm. Food ingestion was measured every 24-h at indicated time. Each experiment included an identical, CAFE chamber without flies to determine evaporative losses which were subtracted from experimental readings. Data were analyzed with LM using JMP.

## $H_2O_2$ stress assay

For $H_2O_2$ stress assays, 7-day old flies were transferred onto 5% sugar and agar food containing 5% hydrogen peroxide (Sigma). Data were collected and analyzed by *log-rank test* using Excel (template available at http://piperlab.org/resources/).

## qPCR

RNA and DNA from same samples of whole flies, guts or brains or 3-day larvae were isolated using TRIZOL (Invitrogen) according to manufacturer's instructions and RNA converted to cDNAs using random hexamers and Superscript II (Invitrogen). Extracted total DNA concentration was measured by NanoDrop Spectrophotometer (Thermo Fisher). qPCR was performed on an Applied Biosystems QuantStudio 6 Flex real-time PCR instrument with Power SYBR Green PCR Master Mix (ABI), relative to a standard curve. Actin5c or total DNA isolated from same samples were used for normalization in expression analysis. Dissection-batch effect was removed by scaling data from each batch with the batch mean, values were expressed relative to a control and the data analyzed with a LM in JMP.

Primers used: dMaf1 forward, ACAAACAAGTGGCATCACGA; dMaf1 reverse CAGCCAAAGTTTGTGGAGGT; pre-tRNA$^{His}$ forward, CGTGATCGTCTAGTGGTTAG; pre-tRNA$^{His}$ reverse, CCCAACTCCGTGACAATG; pre-tRNA$^{Ile}$ forward, CGCACGGTACTTATAATCAG; pre-tRNA$^{Ile}$ reverse, CCAGGTGAGGCTCGAACTC; pre-tRNA$^{Leu}$ forward, GCGCCAGACTCAAGATTG; pre-tRNA$^{Leu}$ reverse, TGTCAGAAGTGGGATTCG; pre-tRNA$^{Tyr}$ forward, GAGCGGTGGACTGTAGAAGTT; pre-tRNA$^{Tyr}$ reverse, CCTATGGATGCCTGCTGGATT; Polr3A forward, CTGCAGCAATGATTCCTG; Polr3A reverse, GTGCCCGAACCCATTGTG; Polr3D forward, GGGTGACCCAGAGTCCCT; Polr3D reverse, GGCGAGCTCAGCGAAGAG; GTF3C3 forward, AACAGGTCATCATGGGTCCAC; GTF3C3 reverse, GTCCACCCTCTTGGCATCC; GTF3A forward, TTGCGACAAGACGTACTCGAT; GTF3A reverse, GGGTCATGTTGCTAACCGAGA; Pbp95 forward, GGGCACGTTTCGCTTCAAG; Pbp95 reverse, CGTAATCCTCGTTGTTAGGACAG; 5S rRNA forward, GCCAACGACCATACCACGCTG; 5S rRNA reverse, AGTACTAACCGCGCCCGACG; Actin5C forward, CACACCAAATCTTACAAAATGTGT; Actin5C reverse, AATCCGGCCTTGCACATG.

## Puromycin incorporation assay and western blotting

Brains of female flies of indicated age and from indicated treatment groups were dissected in ice-cold PBS, with wild-type flies from SYA food processed in batches carried out in parallel. Brains were collected in 200 µl of cold Schneider's medium (Sigma, #S0146) were immediately transferred to a further 800 µl of prewarmed (25°C) Schneider's medium supplemented with 10 µg of puromycin (GIBCO, #A1113803) and incubated at 25°C for 30 min. Reactions were stopped by adding 333 µl of ice-cold 50% trichloroacetic acid and then placed on ice. Samples were homogenized by microtube pellet pestle rods and centrifuged for 15 min at 14,000 rpm at 4°C. The supernatant was removed, and the pellet washed with 1M Tris base. Twenty µl of Laemmli 2× sample buffer (50% LDS sample buffer, 100 mM DTT, in nuclease-free water) was added to the pellet and the pellet was resuspended at 85°C for 15 min. Protein samples were then separated in NuPAGE 4% to 12%, Bis-Tris Gel (Invitrogen, NP0336BOX) following the manufacturer's instructions, and transferred to a nitrocellulose membrane. Membranes were blocked in 5% skimmed milk (Millipore, 70166) and left overnight in anti-puromycin antibody (Millipore, 12D10) at 1:2,500, or anti-tubulin antibody (Millipore, T6199) at 1:2,000 in 5% skimmed milk at 4°C,

washed and probed with secondary antibodies (1:10,000 dilution) for 1 h at room temperature. Ponceau S (Sigma. 729. Aldrich) staining was used to visualize total protein. The intensity of anti-puromycin staining was quantified from chemiluminescence images (using an Anti-Mouse secondary HRP antibody, Abcam, #ab6789) in Fiji, relative to tubulin. Dissection-batch effect was removed by scaling data from each batch with the batch mean, values were expressed relative to a control and the data analyzed with a LM in JMP.

## Experimental design and statistical analysis

Sample size was determined based on past, similar experiments. Flies were randomly assigned to treatment groups. The investigators were not blinded to the conditions. Flies that escaped or were accidentally killed during lifespan experiments were excluded as censors.

Linear model analyses and ordinal logistic regression were performed in JMP (version 14.3.0), Cox Proportional Hazards in R (version 4.2.3) with the *survival* package, Students *t* test in GraphPad Prism (version 10.1.2). All regression analyses had a fully factorial design. For linear models, *p*-values from *F*-tests are reported. Details of statistical tests are given in figure captions.

## Supporting information

**S1 Fig. Loss of *dMaf1* cannot rescue the detrimental effects of a high-sugar diet on lifespan. A**, Lifespan of females with adult-specific, ubiquitous induction of $dMaf1^{RNAi(V109142)}$ driven by *Da-GS* fed a high sugar diet ($-$RU486: $n = 130/20$, +RU486: $n = 138/12$, $p = 0.0652$, *log-rank test*). **B**, Lifespan of females with adult-specific, ubiquitous induction of $dMaf1^{RNAi(V109142)}$ driven by *Act5C-GS* fed a high sugar diet ($-$RU486: $n = 148/2$, +RU486: $n = 136/14$, $p = 3 \times 10^{-4}$, *log-rank test*). These experiments were done at the same time as lifespans in Fig 1. Data underlying the graphs in this figure can be found in S1 Data.
(TIF)

**S2 Fig. Pan-neuronal *dMaf1* knockdown decreases *dMaf1* mRNA level in brains and consistently extends lifespan. A**, qPCR quantifications normalized by *Actin5C* mRNA expression of pre-tRNAs in female brains ($n = 6$ biologically independent samples, age effect, $p < 1 \times 10^{-4}$, pre-tRNAs effect, $p = 0.7430$, age-by-pre-tRNAs interaction, $p = 0.5198$, *LM*), or guts ($n = 6$ biologically independent samples, age effect, $p = 0.4169$, pre-tRNAs effect, $p < 1 \times 10^{-4}$, age-by-pre-tRNAs interaction, $p < 1 \times 10^{-4}$, *LM*) from same wild-type flies with 7- and 42-days' age. **B**, qPCR quantification of *dMaf1* mRNA in female brains after RU486 induction of $dMaf1^{RNAi(V109142)}$ under *Elav-GS$^{Tricoire}$* ($n = 4$ biologically independent samples, $p < 1 \times 10^{-4}$, *Student t test*). **C**, **D**, The second trail of the lifespan assay ($-$RU486: $n = 147/2$, +RU486: $n = 132/1$, $p = 8.85 \times 10^{-5}$, *log-rank test*), and the third trail ($-$RU486: $n = 154/2$, +RU486: $n = 132/17$, $p = 3.65 \times 10^{-7}$, *log-rank test*) on flies with adult-specific, pan-neuronal induction of $dMaf1^{RNAi(V109142)}$ driven by *Elav-GS$^{Tricoire}$*. **E**, Lifespan of females with adult-specific, pan-neuronal induction of $dMaf1^{RNAi(V109142)}$ driven by *Elav-GS$^{301}$* ($-$RU486: $n = 151/3$, +RU486: $n = 142/6$, $p = 6.86 \times 10^{-5}$, *log-rank test*). **F**, Lifespan of females with adult-specific, pan-neuronal induction of $dMaf1^{RNAi(V109142)}$ driven by *nSyb-GS* ($-$RU486: $n = 174/0$, +RU486: $n = 172/2$, $p = 1.68 \times 10^{-9}$, *log-rank test*). **G**, qPCR quantification of *dMaf1* mRNA ($n = 4$ biologically independent samples, $p = 0.0026$, *Student t test*) or **H**, pre-tRNAs ($n = 4$ biologically independent samples, genotype effect, $p < 1 \times 10^{-4}$, pre-tRNAs effect, $p = 0.2546$, genotype-by-pre-tRNAs interaction, $p = 0.2546$, *LM*) in female flies expressing $dMaf1^{RNAi(BL64603)}$. **I**, The first trial of the lifespan assay ($-$RU486: $n = 149/4$, +RU486: $n = 139/12$, $p = 2.03 \times 10^{-8}$, *log-rank test*), and **J**, the second trial ($-$RU486: $n = 138/0$, +RU486: $n = 141/0$, $p = 7.59 \times 10^{-5}$, *log-rank test*) of females with adult-specific, pan-neuronal induction of $dMaf1^{RNAi(BL64603)}$ driven by *Elav-GS$^{Tricoire}$*. **K**, Lifespan of males with adult-specific, pan-neuronal induction of $dMaf1^{RNAi(V109142)}$ ($-$RU486: $n = 138/9$, +RU486: $n = 107/41$, $p = 0.0088$, *log-rank test*) or **L**, $dMaf1^{RNAi(BL64603)}$ ($-$RU486: $n = 143/12$, +RU486: $n = 130/19$, $p = 0.0236$, *log-rank test*) driven by *Elav-GS$^{Tricoire}$*.
**M**, Lifespan of females with adult-specific, pan-neuronal induction of *Elav-GS$^{Tricoire}$* ($-$RU486: $n = 140/8$,

+RU486: $n = 151/1$, $p = 0.1727$, *log-rank test*), **N**, *Elav-GS[301]* (−RU486: $n = 114/4$, +RU486: $n = 122/0$, $p = 0.5418$, *log-rank test*), or **O**, *nSyb-GS* (−RU486: $n = 138/2$, +RU486: $n = 144/0$, $p = 0.5418$, *log-rank test*) drivers alone. **P**, Lifespan of females with adult-specific, induction of *dMaf1[RNAi(V109142)]* (−RU486: $n = 144/4$, +RU486: $n = 150/3$, $p = 0.3358$, *log-rank test*) or **Q**, *dMaf1[RNAi(BL64603)]* (−RU486: $n = 112/27$, +RU486: $n = 139/10$, $p = 0.1786$, *log-rank test*) alone. **R**, Survival of female flies with adult-specific, pan-neuronal induction of *dMaf1[RNAi(V109142)]* driven by *Elav-GS[Tricoire]* under $H_2O_2$ stress (−RU486: $n = 150/0$, +RU486: $n = 150/0$, $p = 5.81 \times 10^{-9}$, *log-rank test*). **S**, Survival of female flies with adult-specific, pan-neuronal induction of *Elav-GS[Tricoire]* driver-alone control under $H_2O_2$ stress (−RU486: $n = 90/0$, +RU486: $n = 90/0$, $p = 0.5327$, *log-rank test*). **T**, Lifespan of females with adult-specific, induction of *dMaf1[RNAi(V109142)]* in Dilp2 producing cells driven by *Dilp2-GS* (−RU486: $n = 147/2$, +RU486: $n = 151/1$, $p = 0.2359$, *log-rank test*). **U**, Volume of food consumed by flies with adult-specific, pan-neuronal induction of *dMaf1[RNAi(V109142)]* driven by *Elav-GS[Tricoire]* ($n = 15$ individual flies, RU486 effect on day 1, $p = 0.6518$, RU486 effect on day 2, $p = 0.8412$, *Student t test*) or driver alone ($n = 15$ individual flies, RU486 effect on day 1, $p = 0.9999$, RU486 effect on day 2, $p = 0.8273$, *Student t test*). **V**, Egg laying of females within 24 h at day-7 combined with adult-specific, pan-neuronal induction of *dMaf1[RNAi(V109142)]* under *Elav-GS[Tricoire]*-driven ($n = 10$ biologically independent samples, RU486 effect $p = 0.1972$, *Student t test*) or driver alone ($n = 6$ biologically independent samples, RU486 effect $p = 0.2512$, *Student t test*). **W**, Lifespan of females with adult-specific induction of *dMaf1[RNAi(V109142)]* in fat body and gut driven by S1106-GS (−RU486: $n = 147/3$, +RU486: $n = 133/4$, $p = 0.2929$, *log-rank test*). **X**, Lifespan of females with adult-specific induction of *dMaf1[RNAi(V109142)]* in intestinal stem cells driven by *5961-GS* (−RU486: $n = 140/10$, +RU486: $n = 141/12$, $p = 0.6641$, *log-rank test*). **Y**, Lifespan of females with adult-specific induction of *dMaf1[RNAi(V109142)]* in enterocytes driven by *5966-GS* (−RU486: $n = 148/0$, +RU486: $n = 123/9$, $p = 0.2752$, *log-rank test*). **Z**, qPCR quantifications of pre-tRNAs normalized by *Actin5C* mRNA in female brains with adult-specific, pan-neuronal induction of *dMaf1RNAi[(V109142)]* driven by *Elav-GS[301]* ($n = 12$ biologically independent samples, RU486 effect, $p = 0.0210$, pre-tRNAs effect, $p = 0.8660$, genotype-by-pre-tRNAs interaction, $p = 0.8660$, *LM*),or guts with adult-specific, whole gut induction of *dMaf1RNAi[(V109142)]* driven by *Ti-GS* ($n = 6$ biologically independent samples, RU486 effect, $p < 0.0001$, pre-tRNAs effect, $p = 0.0229$, genotype-by-pre-tRNAs interaction, $p = 0.0229$, *LM*), or fat bodies with adult-specific, fat body induction of *dMaf1RNAi[(V109142)]* driven by $S_1$*106-GS* ($n = 6$ biologically independent samples, RU486 effect, $p < 0.0001$, pre-tRNAs effect, $p = 0.7633$, genotype-by-pre-tRNAs interaction, $p = 0.7633$, *LM*) at day 14. Data underlying the graphs in this figure can be found in S1 Data.
(TIF)

**S3 Fig. *ElavGS* is active in neurons, and *dMaf1* is expressed at low levels within glia. A**, Images from the cell body layer of the central brain show *ElavGS* + RU486 induced nls-GFP expression in REPO-negative (neuronal) cells. **B**, Images from the SCope database of adult *Drosophila* brain reveal *dMaf1* mRNA is highly expressed within neurons but lowly within glia.
(TIF)

**S4 Fig. The effects of an independent *dMaf1* RNAi line on health. A**, Height climbed during negative geotaxis assays by female flies with a later-life (from day 21), pan-neuronal induction of *dMaf1[RNAi(V109142)]* driven by *Elav-GS[301]* ($n = 50$–60 flies, RU486 effect, $p = 0.0011$, age effect, $p < 1 \times 10^{-4}$, RU486-by-age interaction, $p = 0.4625$, *LM*). **B**, Height climbed during negative geotaxis assays by females with adult-specific, pan-neuronal induction of *Elav-GS[Tricoire]* driver alone ($n = 43$–69 flies, RU486 effect, $p = 0.2044$, age effect, $p < 1 \times 10^{-4}$, RU486-by-age interaction, $p = 0.9495$, *LM*), and **C**, later-life (from day 21), pan-neuronal induction of *Elav-GS[301]* driver alone ($n = 43$–69 flies, RU486 effect, $p = 0.1536$, age effect, $p < 1 \times 10^{-4}$, RU486-by-age interaction, $p = 0.1020$, *LM*). **D**, Quantification of day activity ($n = 32$ individual flies, RU486 effect $p = 0.0788$, *Student t test*), **E**, night activity ($n = 32$ individual flies, RU486 effect, $p = 0.1911$, *Student t test*), **F**, day sleep ($n = 32$ individual flies, RU486 effect, $p = 0.1201$, *Student t test*), and **G**, day bouts ($n = 32$ individual flies, RU486 effect, $p = 0.6826$, *Student t test*) of female flies with adult-specific, pan-neuronal induction of *dMaf1[RNAi(V109142)]* driven by *Elav-GS[301]*. **H**, Quantification of night sleep ($n = 32$ individual flies, RU486 effect, $p = 0.3018$, *Student t test*) and **I**, night

bouts ($n = 32$ individual flies, RU486 effect, $p = 0.6056$, *Student t test*) of female flies carrying *Elav-GS*[301] driver alone. **J**, Cumulative proportion of partial and complete smurfs in female flies with carrying *Elav-GS*[Tricoire] driver alone ($n = 170–340$ flies, RU486 effect, $p = 0.8758$, age effect, $p < 1 \times 10^{-4}$, RU486-by-age interaction, $p = 0.9229$, *ordinal logistic regression*). **K**, Height climbed during negative geotaxis assays by female flies with adult-specific, pan-neuronal induction of *dMaf1*[RNAi (BL64603)] driven by *Elav-GS*[Tricoire] ($n = 32–72$ flies, RU486 effect, $p = 3 \times 10^{-4}$, age effect, $p < 1 \times 10^{-4}$, RU486-by-age interaction, $p = 0.0413$, *LM*). **L**, Cumulative proportion of partial and complete smurfs of females with adult-specific, pan-neuronal induction of *dMaf1*[RNAi (BL64603)] driven by *Elav-GS*[Tricoire] ($n = 167–400$ flies, RU486 effect, $p = 0.0117$, age effect, $p < 1 \times 10^{-4}$, RU486-by-age interaction, $p = 0.1310$, *ordinal logistic regression*). Data underlying the graphs in this figure can be found in S1 Data.

(TIF)

**S5 Fig. Loss of neuronal Pol III transcription does not affect lifespan, and gut 5S rRNA or brain *dMaf1* levels do not change during aging, with decreased Pol III subunits levels within aged brains. A**, Lifespan of females with adult-specific, pan-neuronal induction of *Polr3A*[RNAi] driven by *Elav-GS*[Tricoire] (−RU486: $n = 88/1$, +RU486: $n = 92/0$, $p = 0.0606$, *log-rank test*). **B**, Lifespan of females with adult-specific, pan-neuronal induction of *GTF3C3*[RNAi] driven by *Elav-GS*[301] (−RU486: $n = 134/9$, +RU486: $n = 143/0$, $p = 0.3147$, *log-rank test*). **C**, Lifespan of females with adult-specific, pan-neuronal induction of *GTF3A*[RNAi] driven by *Elav-GS*[301] (−RU486: $n = 112/0$, +RU486: $n = 120/0$, $p = 0.3708$, *log-rank test*). **D**, Lifespan of females with adult-specific, pan-neuronal induction of *Pbp95*[RNAi] driven by *Elav-GS*[301] (−RU486: $n = 147/0$, +RU486: $n = 146/4$, $p = 0.2031$, *log-rank test*). **E**, qPCR quantifications of *Polr3A* mRNA normalized to *Actin5C* ($n = 4$ biologically independent samples, $p = 0.0001$, *Student t test*). **F**, qPCR quantifications of *GTF3C3* mRNA normalized to *Actin5C* ($n = 4$ biologically independent samples, $p = 0.0037$, *Student t test*). **G**, qPCR quantifications of *GTF3A* mRNA normalized to *Actin5C* ($n = 4$ biologically independent samples, $p = 0.0096$, *Student t test*). **H**, qPCR quantifications of *Pbp95* mRNA normalized to *Actin5C* ($n = 4$ biologically independent samples, $p = 0.0004$, *Student t test*). E–H mRNA was obtained from 3-day old larvae of the indicated genotype. **I**, Percentage of adults emerged after driving *Polr3A*, *GTF3C3*, *GTF3A*, *Pbp95*, or *RpS5a* RNAi with *Da-gal4* compared to driver alone. **J**, qPCR quantifications of 5S rRNA normalized to total DNA ($n = 4$ biologically independent samples, $p = 0.0018$, *Student t test*) on 3-day old larvae of the indicated genotype. **K**, qPCR quantifications of 5S rRNA in female brains normalized to *Actin5C* mRNA ($n = 6$ biologically independent samples, age effect, $p < 1 \times 10^{-4}$, *Student t test*) and **L**, guts normalized either to total DNA ($n = 6$ biologically independent samples, age effect, $p = 0.5137$, *Student t test*) or **M**, *Actin5C* ($n = 6$ biologically independent samples, age effect, $p = 0.5137$, *Student t test*) from same wild-type flies. **N**, qPCR quantifications of *dMaf1* in female brains normalized to *Actin5C* mRNA ($n = 6$ biologically independent samples, age effect, $p = 0.1722$, *Student t test*). **O**, qPCR quantifications of *Polr3D* and *Polr3A* in female brains normalized to *Actin5C* mRNA ($n = 6$ biologically independent samples, age effect, $p < 1 \times 10^{-4}$, Pol III subunits effect, $p = 0.1848$, age-by-pre-tRNAs interaction, $p = 0.1848$, *LM*). **P**, qPCR quantification of 5S rRNA in brains normalized to *Actin5C* mRNA in 7-day old ($n = 14$ biologically independent samples, RU486 effect, $p = 0.0212$, *Student t test*) and 42-day old ($n = 8$ biologically independent samples, RU486 effect, $p = 0.0493$, *Student t test*) females with adult-specific, pan-neuronal induction of *dMaf1*[RNAi(V109142)] driven by *Elav-GS*[301]. **Q**, qPCR quantification of 5S rRNA in brains normalized to total DNA in 42-day old females with adult-specific, pan-neuronal induction of *GTF3A*[RNAi] driven by *Elav-GS*[301] ($n = 5–6$ biologically independent samples, RU486 effect, $p = 0.0036$, *Student t test*). **R**, qPCR quantification of 5S rRNA in female brains at 42 days after *dMaf1*[RNAi(V109142)] was induced alone or with *GTF3A*[RNAi] by *Elav-GS*[301], relative to total *Actin5C* ($n = 5–6$ biologically independent samples, genotype effect $p = 0.0011$, RU486 effect $p = 0.0085$, genotype-by-RU486 interaction $p = 3 \times 10^{-4}$, *LM*). Data underlying the graphs in this figure can be found in S1 Data.

(TIF)

**S6 Fig. Loss of neuronal *dMaf1* stimulates translation. A**, Ponceau S staining and western blots quantifying puromycin incorporation in female brains of wild-type flies at days 7 and 42. **B**, Ponceau S staining and western blots of

puromycin incorporation in female brains of flies with adult-specific, pan-neuronal induction of $dMaf1^{RNAi(V109142)}$ driven by $Elav\text{-}GS^{301}$ at days 7 and 42. In B, flies of different ages were assayed at the same time.
(TIF)

**S7 Fig.  Loss of neuronal *dMaf1* stimulates translation and loss of neuronal RpS5a does not affect lifespan. A**, Ponceau S staining and western blots quantifying puromycin incorporation in female brains of flies with adult-specific, pan-neuronal induction of $dMaf1^{RNAi(V109142)}$ driven by $Elav\text{-}GS^{301}$ at days 7 and 42. The same cohort of flies was followed though time. **B**, Quantification of puromycin incorporation at 7 days ($n = 4$ biologically independent samples, $p = 0.0015$, *Student t test*). **C**, Quantification of puromycin incorporation at 42 days age ($n = 4$ biologically independent samples, $p = 5 \times 10^{-4}$, *Student t test*). **D**, Lifespan of females with adult-specific, pan-neuronal induction of $RpS5a^{RNAi}$ driven by $Elav\text{-}GS^{301}$ (−RU486: $n = 148/1$, +RU486: $n = 145/3$, $p = 0.4988$, *log-rank test*). **E**, Images from the cell body layer of the female central brain show lysosomes after RU486 induction of $dMaf1^{RNAi(V109142)}$ under $Elav\text{-}GS^{301}$ ($n = 12$ biologically independent samples, $p < 1 \times 10^{-4}$, *Student t test*). **F**, qPCR quantification of *dMaf1* mRNA in female heads after RU486 induction of *36R* under $Elav\text{-}GS^{Tricoire}$ ($n = 4$ biologically independent samples, $p = 0.0168$, *Student t test*). Data underlying the graphs in this figure can be found in S1 Data.
(TIF)

**S1 Data.  Excel file that contains data underlying all graphs.**
(XLSX)

**S1 Raw Images.  Raw, uncropped images of western blot membranes.**
(PDF)

## Acknowledgments

We thank B. Edgar, N. S. Woodling, and T. Niccoli for providing fly stocks; A. Giblin and M. Li for technical assistance; S. Anoar and T. Niccoli for RNA samples (36R heads); J. Labbadia for comments on the manuscript; and the other IHA members for support, comments, and discussion throughout this project. Stocks obtained from the Bloomington Drosophila Stock Center (NIH P40OD018537) were used in this study, and some RNAi lines were made by the TRiP project (Office of the Director R24 OD030002: "TRiP resources for modeling human disease", PI: N. Perrimon). Additional RNAi lines were obtained from the Vienna Drosophila Resource Center.

## Author contributions

**Conceptualization:** Bowen Xu, Linda Partridge, Nazif Alic.

**Investigation:** Bowen Xu, Alexander Hull, Olivia N.M. Hill, Naja Kobal, Enric Ureña.

**Methodology:** Bowen Xu, Linda Partridge, Nazif Alic.

**Supervision:** Linda Partridge, Nazif Alic.

**Writing – original draft:** Bowen Xu, Linda Partridge, Nazif Alic.

**Writing – review & editing:** Linda Partridge, Nazif Alic.

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
