## [Editor Report · Decision Letter 0]

16 Oct 2024

Dear Dr Alic, 

Thank you for submitting your manuscript entitled "Loss of Pol III repressor Maf1 in neurons promotes longevity by preventing the age-related decline in 5S rRNA and translation" for consideration as a Research Article by PLOS Biology.

Your manuscript has now been evaluated by the PLOS Biology editorial staff as well as by an academic editor with relevant expertise and I am writing to let you know that we would like to send your submission out for external peer review.

Once your full submission is complete, your paper will undergo a series of checks in preparation for peer review. After your manuscript has passed the checks it will be sent out for review. To provide the metadata for your submission, please Login to Editorial Manager (https://www.editorialmanager.com/pbiology) within two working days, i.e. by Oct 18 2024 11:59PM.

Kind regards,

Ines

--

Ines Alvarez-Garcia, PhD

Senior Editor

PLOS Biology

---

## [Decision Letter · Decision Letter 1]

17 Dec 2024

Dear Dr Alic,

Thank you for your patience while your manuscript entitled "Loss of Pol III repressor Maf1 in neurons promotes longevity by preventing the age-related decline in 5S rRNA and translation" was peer-reviewed at PLOS Biology. It has now been evaluated by the PLOS Biology editors, an Academic Editor with relevant expertise, and by two independent reviewers. 

The reviews are attached below. You will see that the reviewers find the conclusions novel and interesting, but they have also raised several points that would need to be addressed before we can consider the paper for publication. Reviewer 1 asks for luciferase reporters to assess translation fidelity in flies and also for a comparison of the findings with a recent publication that shows that decreasing protein translation in young adults improved lifespan (10.1038/s41467-023-40618-x). In addition, this reviewer asks for further experiments, including analysing 5S rRNA levels and/or protein synthesis specifically in neurons, rather than brain, and showing if the rescue of 5S rRNA levels in old files brains in the Maf1 knockdown is suppressed by TFIIIA/C knockdown, among others. Reviewer 2 thinks that you should address the apparent sex-specific effects of Maf1 KO more thoroughly, analyse the impact of dMaf1 KD on other physiological parameters, such as agility in male flies, to see if the lifespan increase translates to other aging phenotypes, and consider the possibility that the beneficial effects might result from improved proteostasis capacity. This reviewer also asks to provide more details in the discussion on why increased protein synthesis in neurons has such pronounced, system-wide benefits on lifespan and heathspan, whereas this effect is not replicated in the gut.

The Academic Editor has checked the reviewers' comments and discussed them with us, and we think that not all the requests will be necessary for publication. You should perform the experiment suggested by Reviewer 1 to examine 5S rRNA levels in the double Maf1 GTF3A/C3 knockdowns (point 1) and examine puromycin incorporation from young and old flies +/- Maf1 KD on the same blot so that the differences can be compared directly (point 3). However, regarding the request of using a luciferase reporter to analyse whether the effects on protein synthesis are mediated by changes in translation, while interesting, we won’t require the experiment for publication and, instead you should add some discussion to the manuscript on the possibility that translation fidelity may be important. In addition, you should discuss the differential effects observed in the study from the Pickering lab with reduced translation in young animals, and you should comment on/rebut/edit the text regarding the evidence that Maf1 is neuron specific. We won’t request either that you look at 5S rRNA levels and protein synthesis specifically in neurons (point 2), or to perform an unbiased screening using cell type specific RNAseq to analyse neuronal expression in KD Maf1 vs controls (point 4).

Reviewer 2 asks for more discussion of the differences in effects in male vs female, which is a good suggestion, however we would not make a requirement for publication the suggestion of using RNAseq to compare male vs female effects. You should also clarify whether or not the climbing assay was performed in males as well as in females, or just in one sex, and to report the data if that is the case, but if you focussed on females only, we won’t require that you perform this on males as well.

In light of the reviews, we would like to invite you to revise the work to thoroughly address the reviewers' reports as we indicate. Given the extent of revision needed, we cannot make a decision about publication until we have seen the revised manuscript and your response to the reviewers' comments. Your revised manuscript is likely to be sent for further evaluation by all or a subset of the reviewers.

**IMPORTANT - SUBMITTING YOUR REVISION**

3. Resubmission Checklist

a) *PLOS Data Policy*

b) *Published Peer Review*

Sincerely,

Ines

--

Ines Alvarez-Garcia, PhD

Senior Editor

PLOS Biology

Reviewers' comments

Rev. 1:

In this study, the authors use Drosophila as a model to show that knockdown of Maf1, which represses protein synthesis gene transcription by RNA polymerase III, extends lifespan in part due to restoring age-dependent levels of 5S rRNA, a key component of the ribosome. Their data support that the effect of Maf1 knockdown on lifespan is due to its role in neurons, suggesting that stimulating protein synthesis in aging neurons promotes longevity both through effects within neurons, and neuronal-mediated effects on other tissues including the gut. Loss of Maf1 has previously been reported to extend lifespan in mice and worms, so the novel part of this study is identifying that this functions through restoring age-dependent expression of the 5S rRNA. The positive effect on lifespan by increasing protein synthesis specifically in neurons is an interesting and significant finding because in general, interventions that decrease protein synthesis in the whole organism or in single cell organisms like yeast extend lifespan. Indeed, in S. pombe, loss of Maf1 shows an opposite effect to the phenotype in animals. Although the authors' findings are important, they don't yet go far enough in explaining the mechanism through which enhancing protein synthesis in neurons extends lifespan. In my opinion, it would be important to include some additional experiments to address if enhancements in translation fidelity or protein turnover/autophagy underly the neuronal-specific effects on lifespan observed. Luciferase reporters to assess translation fidelity in flies are available, and have been used to show that enhancing translation fidelity promotes longevity (Martinex-Miguel 2021 Cell metabolism; DOI: 10.1016/j.cmet.2021.08.017). The authors should also compare their findings with a recent study from the Pickering lab (https://www.nature.com/articles/s41467-023-40618-x), which found that decreasing protein translation in young adults actually improved lifespan, which seem to oppose the authors' findings.

In general, the experiments are rigorous with appropriate controls to support the authors' conclusions. However, I have a few points that should be addressed experimentally.

1. 5S rRNA levels were not examined in the double Maf1 GTF3A/C3 knockdowns (Figure 4). Although TFIIIA/C are shown to be necessary for 5S rRNA expression in supplemental data, it is important to demonstrate that the rescue of 5S rRNA levels in the brain of old flies in the Maf1 knockdown (Figure 4F) is suppressed by simultaneous knockdown of TFIIIA/C.

2. The conclusions would also be strengthened by examining 5S rRNA levels and/or protein synthesis specifically in neurons rather than in brain. Although the elav-Gal4 driver used is neuronal-specific, I was not convinced that Maf1 is not expressed in glia because Figure 2A seems to report fairly strong Maf1 mRNA levels in glial cells.

3. Puromycin incorporation in brains (Figure 5B) from young and old flies +/- Maf1 knockdown should be examined in the same blot so that the extent of the protein synthesis rescue in old flies can be assessed directly.

4. Last, although the genetic approaches used provide strong data to support that the 5s rRNA is the key target of Maf1 involved in this lifespan extension, incorporating an unbiased approach to examine neurons from old Maf1 knockdown flies vs controls (eg RNA-seq or similar approach) would be helpful to check that there is not another pathway that becomes induced that could explain the effect on lifespan. For instance, in mice Maf1 knockout increases autophagy.

Rev. 2:

The manuscript presents new findings on the role of protein synthesis in aging in flies. Specifically, the authors investigate the effects of knocking out the gene Maf1, a key regulator that suppresses RNA Polymerase III (Pol III) activity, which is responsible for transcribing rRNAs, tRNAs, and other components essential for ribosome biogenesis. Their results show that neuron-specific Maf1 knockout (KO) in flies extends lifespan and enhances overall fly health. The lifespan extension appears to be associated with increased transcription of 5S ribosomal RNA, elevated ribosomal activity, and enhanced protein synthesis. Notably, this effect seems localized to neurons, as a Maf1 KO in the gut does not produce similar results. These findings are particularly provocative as they challenge the prevailing notion that enhanced protein synthesis is detrimental to longevity—a view widely supported by literature associating reduced protein synthesis with extended lifespan across multiple model organisms. While the authors present compelling data, more transparency is needed in presenting the results, especially for experiments conducted primarily in female flies. Overall, the manuscript is a strong candidate for publication, though some areas require refinement.

The abstract could benefit from greater clarity and simplicity. Some sentences are overly complex, obscuring the main findings. The authors should clearly state that the general knockout of Maf1 extends lifespan predominantly in female flies, while neuron-specific knockdown (KD) shows lifespan extension effects in males as well. The abstract should strictly align with the data presented in the manuscript, with a clear emphasis on female-focused results.

The paper should address the apparent sex-specific effects of Maf1 KO more thoroughly. Both the current study and other cited research suggest that the longevity effect is more pronounced in females, a pattern also observed in mouse models. This is an important biological point that warrants discussion: what makes males less responsive to Maf1 KO than females? Including RNA-Seq data comparing male and female responses to Maf1 KO would add depth, particularly in highlighting any sex-specific molecular pathways that might underlie these differences, and provide a general overview of the molecular pathways affected by the knockdown. This could help elucidate which specific responses are induced and might explain lifespan extension.

Have the authors examined the impact of dMaf1 KD on other physiological parameters, such as agility, in male flies? While the data suggest that males experience a statistically significant, though modest, lifespan increase with dMaf1 KO, it's unclear if this translates to other aging phenotypes. In general, I would suggest that the authors clearly show, at least for some key results, e.g., protein synthesis increase and health benefits in male flies as well. Otherwise, it remains unclear whether the authors performed the experiment (and hid the results) or did not conduct the experiment at all.

The authors should not exclude the possibility that the beneficial effect might result from improved proteostasis capacity. They have stated that mice with dMaf1 knockdown show enhanced autophagy. Previous studies cited by the authors in C. elegans seem to attribute most of the lifespan extension of Maf1 to increased stress resistance. The authors have correctly shown that ribosomal proteins and Pol III are required for the beneficial effect, but it remains important to characterize some of the proteostasis pathways and stress responses to understand how they are modulated and whether they are also necessary for the beneficial effect of Maf1 knockout.

The authors should provide a more detailed discussion on why increased protein synthesis in neurons has such pronounced, system-wide benefits on lifespan and healthspan, whereas this effect is not replicated in the gut. This raises interesting questions about the cell-type specificity of Maf1's role in aging, and a mechanistic explanation for this difference could greatly enhance the discussion.

Minor Points

* Figure 4: If these experiments were performed solely on female animals, this should be clearly stated in the main text as well as in the figure legend.

* Elav-GS System: The authors should provide detailed information on the efficiency and effect of the Elav-GS system for gene suppression, including information for male flies if available.

* Figure S2: P-values should be included directly on the figure panels for clarity and to enhance readability. Additionally, it appears that panels S2B and S2C might be swapped—please verify and correct if necessary.

* Statistical Tests: Clearly indicate the type of statistical tests used and specify results in the figure legends. Any claims of indirect effects should be supported by appropriate citations or clarified in the text.

---

## [Decision Letter · Decision Letter 2]

9 May 2025

Dear Dr Alic,

Thank you for your patience while we considered your revised manuscript entitled "Loss of Pol III repressor Maf1 in neurons promotes longevity by preventing the age-related decline in 5S rRNA and translation" for publication as a Research Article at PLOS Biology. This revised version of your manuscript has been evaluated by the PLOS Biology editors, the Academic Editor and the two original reviewers.

Based on the reviews (attached below), we are likely to accept this manuscript for publication, provided you satisfactorily address the remaining points raised by Reviewer 2. Please also make sure to address the data and other policy-related requests stated below.

We expect to receive your revised manuscript within two weeks. 

*Published Peer Review History*

*Press*

Sincerely,

Ines

--

Ines Alvarez-Garcia, PhD

Senior Editor

PLOS Biology

DATA POLICY:

Thank you for submitting the data underlying the graphs shown in the figures. Please also ensure that figure legends in your manuscript include information on where the underlying data can be found. Also, please name the file F

Please ensure that all data files are invariably referred to (in the manuscript, figure legends, and the Description field when uploading your files) using the following format verbatim: S1 Data, S2 Data, etc. Also, please save the excel file using exactly the following convention: S1_Data.xlsx (using an underscore).

We require the original, uncropped and minimally adjusted images supporting all blot and gel results reported in an article's figures or Supporting Information files. We will require these files before a manuscript can be accepted so please prepare and upload them now. Please carefully read our guidelines for how to prepare and upload this data: https://journals.plos.org/plosbiology/s/figures#loc-blot-and-gel-reporting-requirements

CODE POLICY

Reviewers' comments

Rev. 1:

The authors have carefully addressed our concerns and the addition of the new control data in Figures 4 and 5 provides stronger support for their model. We also think the additional text in the discussion helps to better describe their work in the context of other related papers. We have no additional concerns.

Rev. 2:

In this revised manuscript, Xu and colleagues have introduced significant corrections and clarifications, which substantially strengthen the manuscript's overall quality and suitability for publication. In particular, the clarifications concerning the sex of the individuals studied, as well as the expanded discussion section, contribute to a more objective and precise presentation of the findings. I acknowledge that conducting additional experiments would have considerably delayed the publication process; nevertheless, the manuscript represents a strong candidate for the proposed journal. I would recommend addressing a few remaining minor issues prior to final acceptance.

I believe the abstract still requires significant improvement for clarity. As currently written, it is not immediately clear which statements refer to previously established knowledge and which describe the novel findings of the present study. Including a sentence such as 'In this work, we show that…' would greatly enhance readability and help distinguish the authors' contributions. For instance, the sentence 'Knock-down of Maf1 extends lifespan' is ambiguous—it is unclear whether this refers to findings presented in this manuscript or to previously published results in other model organisms. If this is a novel result from the current study, it should be explicitly stated.

Regarding the assessment of autophagy, I recommend a more cautious phrasing of some of the conclusions. As LysoTracker stains acidic compartments such as lysosomes, it does not, by itself, provide information about the dynamic process or flux of autophagy. Rather, it reflects the size and number of acidic/lysosomal compartments, which may—though not necessarily—be associated with increased proteome turnover or autophagic activity. The authors should clarify in the text that their measurements pertain to lysosomal compartment number and size, which can indirectly suggest, but do not directly demonstrate, altered autophagy.

Aside from these comments, I find the study to be of interest and believe it will make a valuable contribution to the field of aging biology. I recommend publication following the implementation of the suggested revisions.

---

## [Editor Report · Decision Letter 3]

6 June 2025

Dear Dr Alic,

Thank you for the submission of your revised Research Article entitled "Loss of Pol III repressor Maf1 in neurons promotes longevity by preventing the age-related decline in 5S rRNA and translation" for publication in PLOS Biology. On behalf of my colleagues and the Academic Editor, Josh Dubnau, I am delighted to let you know that we can in principle accept your manuscript for publication, provided you address any remaining formatting and reporting issues. These will be detailed in an email you should receive within 2-3 business days from our colleagues in the journal operations team; no action is required from you until then. Please note that we will not be able to formally accept your manuscript and schedule it for publication until you have completed any requested changes.

PRESS

Sincerely, 

Ines

--

Ines Alvarez-Garcia, PhD

Senior Editor

PLOS Biology
